


**High-resolution simulation of link-level vehicle emissions and**
**concentrations for air pollutants in a traffic-populated East Asian city**
Shaojun Zhang [1, 2], Ye Wu [1, 3 *], Ruikun Huang [1], Han Yan [1], Yali Zheng [1, 4], Jiming Hao [1, 3]
1 School of Environment and State Key Joint Laboratory of Environment Simulation and Pollution
Control, Tsinghua University, Beijing 100084, China, Tsinghua University, Beijing 100084, P. R. China
[2] Department of Mechanical Engineering, University of Michigan, Ann Arbor, MI 48109, U.S.
[3] State Environmental Protection Key Laboratory of Sources and Control of Air Pollution Complex,
Beijing 100084, P. R. China
[4] Society of Automotive Engineers of China, 102 Lianhuachi East Road, Beijing 100055, P.R. China
*Correspondence to:* Y. Wu (ywu@tsinghua.edu.cn)
*Abstract:*
Vehicle emissions of air pollutants created substantial environmental impacts on air quality for many
traffic-populated cities in East Asia. A high-resolution emission inventory is an irreplaceable tool
compared with traditional tools (e.g., registration data based approach) to accurately evaluate real-world
traffic dynamics and their environmental burden. In this study, Macao, one of the most populated cities
in the world, is selected to demonstrate a high-resolution simulation of vehicular emissions and their
contribution to air pollutant concentrations by coupling multi-models. First, traffic volumes by vehicle
category on 47 typical roads were investigated during weekdays of 2010 and further applied in a
networking demand simulation with the TransCAD model to establish hourly profiles of link-level
vehicle counts. Local vehicle driving speed and vehicle age distribution data were also collected in
Macao. Second, based on a localized vehicle emission model (e.g., the EMBEV-Macao), this study
established a link-based vehicle emission inventory in Macao with high resolution meshed in a temporal
and spatial framework. Furthermore, we employed the AERMOD model to map concentrations of CO,
$NO_2$ and primary $PM_{2.5}$ contributed by local vehicle emissions during the weekdays of November 2010.
This study has discerned the strong impact of traffic flow dynamics on the temporal and spatial patterns
of vehicle emissions, such as a geographic discrepancy of spatial allocation up to 25% between THC
and $PM_{2.5}$ emissions owing to spatially heterogeneous vehicle-use intensity between motorcycles and
diesel fleets. We also identified that local vehicles are a dominant source of ambient $NO_2$ in





traffic-populated areas as evidenced by good agreement between AERMOD-simulated data and
observed results. Therefore, this paper provides a case study and a solid framework for developing
high-resolution environment assessment tools for other vehicle-populated cities in East Asia.
**1. Introduction**

6        The soaring vehicle stock driven by social-economic development has created a series of

substantial challenges regarding air pollution, energy insecurity, and public health within many countries
(Uherek et al., 2010; Saikawa et al., 2011; Shindell et al., 2011; Walsh, 2014). At the national level, we
take nitrogen oxides ($NO_X$) emissions as an example as it is an essential precursor to the formation of
ozone and nitrate aerosol in the atmosphere. On-road vehicles are currently responsible for 29% of
national anthropogenic $NO_X$ emissions in China (MEP, 2014), 37% in U.S. (U.S. EPA, 2014) and 40%
in Europe Union (EEA, 2014; Vestreng et al., 2009). At the city level, the vehicular contribution to
ambient nitrogen dioxide ($NO_2$) concentration is very significant in traffic related areas (Carslaw et al.,
2011). For example, in European countries where diesel vehicles make up a considerable part of private
passenger cars, near-road $NO_2$ concentration exceeds the ambient air quality standard. This issue is seen
as one of the most significant air pollution problems in Europe although great efforts have been made to
cope with the $NO_2$ exceedance, including the implementation of stringent emission standards for diesel
vehicles (e.g., the latest Euro 6 requirements) (Franco et al., 2014; Carslaw et al., 2011; Carslaw and
Rhys-Tyler, 2013; Chen and Borken-Kleefeld, 2014). Higher health risk as a result of exposure to
vehicular emissions (e.g., particle, $NO_X$) is understandable in traffic-populated cities, and is probably
associated with the large resident population, greater traffic congestion and unfavorable dispersion due
to dense buildings (Du et al., 2012; Ji et al., 2012). In 2012, the International Agency for Research on
Cancer Group 1 assessed the carcinogenicity of diesel emissions as "carcinogenic to humans" with
sufficient evidence for it to be characterized as a cause of lung cancer (Benbrahim-Tellaa et al., 2012).

25       The high-resolution vehicle emission inventory is an irreplaceable tool to accurately evaluate

impacts on air quality and public health, as it can well reflect the close connections between
environmental impacts and traffic flows. McDonald et al. (2014) analyzed the impacts of enhanced
spatial resolution from 10 km to 500 m on vehicular $CO_2$ emission inventory for Los Angeles, which
clearly demonstrated substantial improvements in the accuracy for areas containing traffic-dense
microenvironments (e.g., heavily trafficked highways). Consequently, link-based emission inventory is a
preferred tool owing to its substantial advantage in spatial resolution for local traffic and environmental





management. Over the past decade, high-resolution emission inventory initiatives have been carried out in China's vehicle-populated cities. Taking Beijing, the capital city of China for example, Huo et al. (2009) established a link-based emission inventory for light-duty gasoline vehicles (LDGVs) in the urban area based on estimated emission factors with the IVE model. However, significant emissions of $NO_X$ and fine particulate matter ($PM_{2.5}$) may be attributed to heavy-duty diesel vehicles (HDDVs) instead of LDGVs, including the gross emitters registered in other provinces (Wang et al., 2011 and 2012a), whose contributions are currently not evidenced in the registration-based inventories for China's vehicle-populated cities (Wu et al., 2011; Zhang et al., 2014a; Zheng et al., 2014). Wang et al. (2009) and Zhou et al. (2010) estimated vehicular emissions for the urban area of Beijing by using grid-based data of average speed and aggregated vehicle kilometers travelled. However, their resolutions are not sufficient to present hourly fluctuation of network traffic volume and quantify vehicular emissions at the link level.

As traffic management actions become more important for vehicle emission control, such as the license control policies effective in seven vehicle-populated cities of China (e.g., Shanghai, Beijing, Guangzhou, Tianjin, etc.) and the Electronic Road Pricing (ERP) program adopted in Singapore (Goh, 2002). We therefore envision greater demand for high-resolution vehicle emission inventories by local environmental protection administrations in the near future. A few technical barriers are expected to be shortly overcome for improving the high-resolution vehicular emission inventory based on the development experience of the London Atmospheric Emission Inventory (LAEI) (TfL, 2014). First, high-resolution traffic data including traffic counts, vehicle speed and fleet composition should be investigated or estimated at the link level with hourly fluctuations. Second, real-world emission factors should be developed based on a sufficient measurement database to effectively address potential uncertainties (e.g., gaps between regulatory cycle and off-cycle conditions) (Carslaw et al., 2011; Wu et al., 2012; Zhang et al., 2014a). Third, technology allocations of the total fleet (e.g., traffic counts by fuel type and vehicle age) should be derived based on real-world traffic data instead of registration data, considering vehicular emissions are fairly sensitive to vehicle technology allocations (Vallamsundar and Lin, 2012). Finally, the application of high-resolution emission inventory can be significantly enhanced by extending the evaluation framework from vehicular emissions to pollutant concentration, which are of overriding concerns to residents, pedestrians and policy-makers (Vallamsundar and Lin, 2012; Misra et al., 2013).



In this study, we selected Macao as a case city to demonstrate high-resolution simulation for
vehicle emissions and primary concentrations of air pollutants in this traffic-populated city. Macao is
well-renowned for its tourism and gaming industry, which attracts numerous visitors and created huge
transportation demand. Owing to the absence of massive rail-based public transit system, which is now
under construction in Macao, local transportation completely depends on on-road vehicles. The
vehicle-population density (including motorcycles, MCs) in Macao is approaching 7800 veh km$^{-2}$ in
2014, significantly more dense as compared with other East Asian cities (e.g., 430 veh km$^{-2}$ of Shanghai,
340 veh km$^{-2}$ of Beijing and 700 veh km$^{-2}$ of Hong Kong) (DESC, 2014; HKS, 2014; NBSC, 2014).
Furthermore, Macao's total vehicle population has surpassed 240 thousand in 2014, more than double
the level in 2000 (DESC, 2014). Significant gridlock has been caused due to rapid motorization in the
Macao Peninsula during rush hours, when the average speed of arterial roads is frequently lower than 15
km h$^{-1}$ (TMB, 2010). On the other hand, local air quality data indicate several nonattainment sites for
annual ambient $PM_{2.5}$ and $NO_2$ concentrations in the traffic-dense and residential areas of Macao (DESC,
2014). On-road vehicles have been identified as the major local contributor to air pollution, because
industrial emissions in Macao are quite minor compared with the on-road transportation sector. Thus,
there is an urgent need to attach importance to controlling vehicular emissions with the support of
high-resolution emission inventory technology in this traffic-populated city.
**2. Methodology and data**
**2.1 General study framework and components**
This study generally consists of three components: (1) characterizing hourly traffic profiles at the
link level, (2) establishing a high-resolution vehicle emission inventory, and (3) simulating the
concentrations of major air pollutants contributed by local vehicle emissions in Macao (see Fig. 1). The
core task of this study is to calculate emissions of air pollutants and carbon dioxide ($CO_2$) from local
vehicles meshed in the high resolution matrix of the "hour-link-vehicle technology group", which is
illustrated by Equation 1.
$$E_{h, l, p, v} = \sum_{f, y} 10^{-3} \cdot EF_{f, p, v, y} \cdot L_l \cdot TV_{h, l, v} \cdot VF_{f, v, y} \quad (1)$$
where $E_{h, l, p, v}$ are the emissions of pollutant category p from vehicle classification v during hour h for
link l, kg h$^{-1}$; $EF_{f, p, v, y}$ is speed-dependent average emission factor of pollutant category p for vehicle
technology group defined by classification v, fuel type f and vehicle age y, g veh$^{-1}$ km$^{-1}$; $L_l$ is the total





length of link l, km; $TV_{h, l, v}$ is total traffic volume of vehicle classification f during hour h, veh h$^{-1}$; and
$VF_{f, v, y}$ is the volume fraction of vehicle technology group defined by fuel type f and vehicle age y. We
define eight vehicle classifications in this study that were recognized from road traffic video records as
follow: light-duty passenger vehicle (LDPV), MC, taxi, public bus (PB), medium-duty passenger vehicle
(MDPV), heavy-duty passenger vehicle (HDPV), light-duty truck (LDT) and heavy-duty truck (HDT).

6        Therefore, we further characterized total hourly emissions from the total vehicle fleet based on the

bottom-up method, namely from each link to the entire road net, as Equation 2 illustrates.

8        $$E_{h, p} = \sum_{l, v} E_{h, l, p, v} \qquad (2)$$

where $E_{h, p}$ are the total vehicle emissions of pollutant category p during hour h from the total vehicle
fleet in Macao, kg h$^{-1}$. In the following two sub-sections, we present detailed methods for developing
high-resolution traffic data and vehicle emission factors. Due to the time limitation on the traffic field
investigation, we only focus the case study for weekdays during 2010; weekends were not investigated
when traffic flows might be different.
**2.2 Summary of geography and road network in Macao**

16       Macao is one of the two Special Administrative Regions (SAR) in China lies on the western side

of the Pearl River Delta, with a total land area of only 30 km$^2$, which is the most densely populated city
in the world (~20 thousand people km$^{i2}$) (DSEC, 2014). The Macao SAR now consists of the Macao
Peninsula (MP) and the Taipa-Cotai-Coloane (TCC) islands (See Fig. S1). In particular, the CoTai
Reclamation Area is a piece of newly reclaimed land on the top of the bay area between Taipa and
Coloane, where new casinos and hotels have been constructed since land of Macao is scare. Nearly 90%
of Macao's total population is concentrated in the MP, where the population density is significantly
higher than the combined density of Taipa-CoTai-Coloane (TCC) regions (i.e., 54 thousand vs. 4.3
thousand, unit in people km$^{-2}$). The MP geographically consists of five regions, nominally parishes.
Among those five parishes, the St. Anthony Parish where the Ruins of St. Pual's Cathedral is located has
the highest population density, which is approaching 120 thousand people km$^{-2}$.

27       Based on the GIS database of road network in Macao provided by the Macao Transportation

Bureau, there were a total of 1704 road links in the study year of 2010. We categorized all those links
into three road classes: urban freeways, arterial roads and residential roads, representing that the level of
service decreasing from high to low. It should be noted that the road links are unevenly distributed





among various areas of Macao, but similar to the spatial patterns. For example, 77% of all road links
(i.e., 1306 links) were concentrated in the Macao Peninsula, which were responsible for 59% of Macao's
total road length.
**2.3 Field investigation and simulation of link-based traffic data**
We investigated traffic data on 47 typical road links during three field investigation periods from
Jan 2010 to Jan 2011 (i.e., nearly 20 weekdays during Jan 2010, May 2010 and Jan 2011), according to
the spatial heterogeneity of road network in Macao by covering all road classes and regions. The real
traffic flow records of each link was collected with a portable video camera for at least 20 minutes
within each hour. Among all links investigated, 5 typical road links were investigated for the entire day
(i.e., 24-h sampling). Sampling duration for the rest of the links investigated in general were from 6 a.m.
to 11 p.m. (i.e., day-time sampling). Detailed hourly traffic volumes by vehicle classification for 47 road
links were further broken down based on those original video profiles by major region and road class
(see Table 1). We can clearly observe variations in hourly total traffic counts for three road classes, with
significant peaks of traffic demand during morning and evening rush hours (see Fig. 2 and Table 1).
Traffic volume fraction by vehicle classification is another essential type of data obtained from
traffic video record (see Fig. S2 as an example of arterial roads). During the evening rush hour (6 p.m.),
LDPVs and MCs contributed nearly 80% of total traffic volume, which are the two major vehicle types
used for daily commuting demand in Macao. In particular, MCs are low-cost commuting vehicles for the
relatively lower income group in Macao. Therefore, the observed traffic fraction of MCs was higher
than that of LDPVs on arterial roads of the Macao Peninsula. By contrast, observed traffic fraction of
MCs in the TCC was only approximately 15%. In addition to the spatial variations among various road
classes and areas, we also observed temporal variations of various vehicle classifications. Taking arterial
roads in the MP for example, their average traffic fractions of taxis were approximately 10% during the
day time (6 a.m. to 12 p.m.). During the night time (12 p.m. to 6 a.m.), accompanied by significantly
reduced traffic demand of MCs and LDPVs, taxis could be responsible for 20~30% of total vehicle
counts. Due to the minor economic contribution of local industry, the average traffic fraction of trucks in
Macao indicating freight transportation was significantly lower than those in Beijing and Guangzhou.
The TransCAD 5.0 model was applied to estimate total traffic demand and its spatial allocation at
the link level. TransCAD 5.0, one of the most widely-used traffic planning software, can estimate
origin-destination (OD) matrix of the road network from link traffic counts. In this study, we selected





the multiple path matrix estimation (MPME) procedure provided by the TransCAD 5.0 and estimated
total traffic volumes of all road links during the 6 p.m. hour with observed hourly traffic counts of 33
links as input data. After a number of iteration runs, the average discrepancy between simulated traffic
volumes and the observed values (i.e., output vs. input) is 4.3% and the Pearson coefficient is 0.95,
indicating statistically satisfactory results (see Fig. S3). For other hours, we estimated hourly total traffic
volumes based on the averaged temporal allocations and simulated traffic volumes during the 6 p.m.
hour, as Equation 3 illustrates.
$$TV_{h,1} = TV_{18,1} \cdot \frac{\overline{\alpha_{a,c,h}}}{\overline{\alpha_{a,c,18}}} \qquad (3)$$
where $TV_{h,1}$ is the hourly total traffic volume for road link l during the hour h, veh h$^{-1}$, and $TV_{18,1}$ is
particularly the hourly data during the 6 p.m. hour simulated by the TransCAD if observed traffic
volume data is unavailable); $\overline{\alpha_{a,c,h}}$ is the averaged ratio of hourly total traffic volume during the hour h
to daily total traffic volume for the area a and the road class c. Therefore, the traffic volumes by vehicle
classification are further estimated based on the traffic fraction data averaged by area, road class and
hour.
In addition to traffic volume, traffic condition indicated by link-based hourly speed is another
category of essential input data. First, we used a portable GPS receiver to collect second-by-second
vehicle trajectory data for on-road vehicles during the same field sampling periods of traffic counts.
Considering the distinctions of driving behaviors among MCs, PBs and other vehicle classifications (e.g.,
passenger vehicles and trucks), like more frequent stops for PBs to discharge and receive passengers, we
used a taxi equipped with the GPS receiver to chase LDPVs randomly to represent traffic conditions for
on-road vehicles other than PBs and MCs. Each targeted vehicle was chased for at least 10 minutes. For
PBs and MCs, we selected typical vehicles to record their traffic trajectory data. In this study, we
collected traffic trajectory data of LPDVs, PBs and MCs for 115 thousand seconds, 86 thousand seconds
and 30 thousand seconds, respectively, with high abundance of spatial and temporal distribution. Second,
we integrate the original second-by-second GPS trajectory data with the road network GIS system to
identify the road link information (e.g., link name, parish and road class) for each sampling second.
Third, we estimated averaged hourly speed for each road class in each parish. To validate the speed
profiles, we observed variations in average hourly speeds by area and road class for LDPVs as an
example, which were aggregated by link-level speed profiles with traffic volume data taken into account





(see Fig. 3). Clearly, average hourly speeds for arterial and residential roads in the MP were lower than
20 km h$^{-1}$ for longer than 15 hours (e.g., from 6 a.m. to 8 p.m.), indicating extremely congested traffic
conditions. In particular, average hourly speeds during the evening rush period (e.g., 6 p.m. and 7 p.m.)
were even less than 15 km h$^{-1}$, which corresponded to the officially released data. In the TCC, where
traffic is less populated, average hourly speeds for arterial and residential roads were significantly higher
than those in the Macao Peninsula, ranging from 20 km h$^{-1}$ to 40 km h$^{-1}$ except for the 6 p.m. hour. On
the other hand, we could also observe differences of aggregated daily speed among various vehicle
classifications (see Fig. S4). For example, average daily speed of taxis was 24.0 km h$^{-1}$, higher than the
21.7 km h$^{-1}$ of LDPVs, due to higher traffic volume fraction of taxis in the night time when there were
usually free traffic flows. Similarly, average speed of HDTs was 27.0 km h$^{-1}$, topping all vehicle
classifications, because their traffic volume fraction was significantly higher in the TCC compared to the
MP.
**2.4 Emission factor development and the integration with traffic data and vehicle age distribution**

15        We initiated a comprehensive measurement program of collecting real-world emission profiles

since 2010, in order to establish and update a localized emission factor model for vehicles in Macao (e.g.,
the EMBEV-Macao model). So far, more than 60 typical vehicles, LDPVs, taxis, PBs, LDTs and HDTs,
have been measured on road by using a portable emission measurement system (PEMS). Furthermore, a
large-scale remote sensing vehicle emission measurement project was conducted during March and
April 2008, which enabled the collection of fuel-based emission factors for MCs in Macao. Detailed
experimental section in Macao and the measurement results are documented in several of our previous
papers regarding gasoline, diesel and more advanced vehicles (e.g., hybrid electric vehicles) (Hu et al.,
2012; Wang et al., 2014; Zhang et al., 2014b; Zhou et al., 2014; Wu et al., 2015a and 2015b; Zheng et
al., 2015). We developed an emission factor model, the EMBEV-Macao model, with reference to the
modeling framework and methodology of the EMBEV model which is originally developed for the
vehicle fleet in Beijing (Zhang et al., 2014a). Technically, we used local measurement data to estimate
basic emission factors of air pollutants and $CO_2$ emissions under their typical driving conditions by
vehicle age group. Second, we developed localized speed correction curves based on a micro-trip
method for each vehicle classification to integrate vehicle emission factors and traffic conditions at the
link-level (Zhang et al., 2014b and 2014c; Wu et al., 2015). Furthermore, the EMBEV-Macao model
enables us to correct impacts of local temperature, fuel quality, air conditioning usage, and other aspects



to the real conditions. For example, the sulfur content of gasoline and diesel were approximately 90 ppm
and 15 ppm during 2010.
Considering that there was no significant policy influencing traffic flow composition during
2008-2010, we estimated detailed traffic fraction by fuel type and vehicle age for each vehicle
classification based on the vehicle information database from the 2008 remote sensing project (Zhou et
al., 2014). It should be noted that some vehicle classifications have a single fuel type; e.g., gasoline for
MCs and diesel for PBs. By contrast, other vehicle specifications like engine displacement have a more
important effect on real-world emissions. Therefore, we also derived the on-road traffic volume split
ratios by engine displacement for MCs and PBs (refer to the footnote of Table 2). Table 2 illustrates the
detailed traffic volume fraction by vehicle age and fuel type (or split by engine displacement for MCs
and PBs) for each vehicle classification.
**2.5 Modeling dispersion of vehicular air pollutants**
Urban air quality models are commonly used to estimate the spatial distribution of vehicular
pollutants by simulating their chemical and physical processes in the atmosphere within urban areas.
Holmes and Morawska (2006) classified dispersion models into Box models, Gaussian models,
Lagrangian models, Computational Fluid Dynamic (CFD) models. Currently, Gaussian models are
recommended by the environmental protection agency of most countries all over the world.
The AMS/EPA regulatory model (AERMOD) is a steady state Gaussian plume dispersion model
which is recommended by U.S. EPA (U.S. EPA, 2004). The modeling system consists of one main
program (AERMOD) and two pre-processors (i.e., AERMET and AERMAP). In addition, calculating
urban boundary layer parameters and considering urban heat island effect makes AERMOD sensitive for
local meteorological conditions. Recently, several studies have investigated the integration performances
of the traffic simulation model, vehicle emission model and the AERMOD model. For example,
Vallamsundar and Lin (2012) integrated MOVES and AERMOD models to simulate the $PM_{2.5}$ hotspot
cases of typical roads in U.S. cities (i.e., study domain area of ~0.5 $km^2$) and provided some
implications based on sensitivity analysis, such as narrowing the data gap between traffic, emissions and
air quality models and further investigation of important local input data (e.g., traffic composition, fleet
age distribution). Misra et al. (2013) also integrated a traffic simulation model, a vehicle emission model
and the AERMOD model to estimate traffic-related pollution in downtown Toronto (i.e., study domain
area of ~0.5 $km^2$). It should be noted that, in those previous investigations at near-field level (Zannetti,





1990), the AERMOD simulated vehicular emissions as a series of point sources which approximate a
traffic lane.
Considering a significantly larger study area, higher road density and the scarcity of metrological
data and surrounding building profiles in a sufficiently fine resolution, we divided the study domain into
a grid of 350 square cells (500 m×500 m). Aggregated hourly vehicular emissions of major pollutants
(e.g., CO, $NO_X$ and $PM_{2.5}$) from all road links in each grid are used as the input data for the AERMOD.
The receptors are placed at central points of all cells at a height of 2.0 m. In terms of the geographic data
and the altitude information is obtained from the Google Earth. Building downwash effects are
simulated by the AERMOD. In our study, we model the weekdays of November 2010 when rainy days
were much fewer compared to other months. Hourly meteorological profiles from two monitoring sites
located in MP and TCC respectively, including temperature, wind direction, wind speed, relative
humidity and air pressure are provided by the Department of Metrological Services in Macao. The
northeasterly winds are prevailing during that month, supplemented by a minor part of northerly and
easterly winds (see Fig. S5). Based on the fleet composition and on-road measurement data, we
estimated average volume ratio of primary $NO_2$ to total $NO_X$ emissions as) 10%. In addition, ozone
concentration data from Macao EPB are also used to simulate the oxidation process from freshly emitted
NO to ambient $NO_2$ by the AERMOD.
In order to compare simulated concentrations of air traffic-related pollutants with their ambient
concentrations over the same period (i.e., November 2010), air quality data of local monitoring sites are
provided by the Macao Environmental Protection Bureau (EPB). Furthermore, impacts from regional
background, cross-boundary transport and other area sources are estimated by closing local stationary
(e.g., power and waste incineration plants located in the Coloane Island) and on-road sectors with the
CMAQ model at a spatial-resolution level of 4×4 $km^2$, which add up to 304 μg $m^{-3}$ of CO, 27 μg $m^{-3}$ of
$NO_2$ and 23 μg $m^{-3}$ of $PM_{2.5}$ as the monthly averages. Meanwhile, we employed the AERMOD and
estimated that $NO_2$ concentration contributed by local power and waste incineration plants located in
Coloane were approximately 1 μg $m^{-3}$ in the Coloane Island and more marginal in the MP.

## 28 3. Results and discussion

### 29 3.1 Estimated traffic activity and vehicle emissions

Table 3 presents spatially-explicit traffic counts during a typical weekday and an evening rush
hour (i.e., 6 p.m.), respectively. More than 80% of total daily traffic counts were concentrated in the MP,





160% higher than the overall average of Macao. In particular, the Saint Antony Parish with
internationally-renowned tourist attraction (e.g., the Ruins of St. Paul's) had a top hour-based density of
daily traffic volume as a result of its substantial population density. Furthermore, traffic activity (unit
veh km h$^{-1}$ or veh km d$^{-1}$) can be estimated as the product of traffic counts and link length, namely
$TV_{h, l, v}$  and  $L_l$  (see Equation 1), which is an essential indicator of vehicle-use intensity. Estimated
daily traffic activity of Macao's total vehicles in a typical weekday of 2010 is $4.04 \times 10^6$ veh km d$^{-1}$ (see
Table S1). LDPVs and MCs rank first and second among all vehicle classifications, accounting for 43%
and 30% of total daily traffic activity in Macao. Therefore, fleet-average daily vehicle kilometers
travelled (VKT) of LDPVs and MCs during weekdays of 2010 are 20.8 km and 11.7 km, respectively. If
we ignore potential difference between weekdays and weekends, fleet-average annual VKT of LDPVs
and MCs registered in Macao are 7600 km and 4300 km as of 2010, which are quite comparable with
our previous survey results. Those values could be only responsible for traffic demand within Macao,
considering a part of LDPVs travel cross the boundary of the Macao SAR into Mainland China. It is
worth noting that annual VKT of LDPVs registered in Macao is significantly lower than those of Beijing
and Guangzhou (Zhang et al., 2013 and 2014a). The major reason is the scale of Macao is much smaller
than those megacities of Mainland China (e.g., Beijing, Guangzhou), approximately 15 km from the
northernmost parish in MP to the Coloane Island. Since fewer MCs drive on the cross-sea bridges, a
major part of MCs' traffic activity (note: in particular for light-duty two-stroke MCs) is largely limited
within MP or TCC. Therefore, traffic activity of MCs is lower than LDPVs although with higher traffic
counts, whose estimated annual VKT is comparable to the value in Mainland China (e.g., 5000~6000
km) (Zhang et al., 2013 and 2014a).

22        Table 4 presents estimated average distance-specific emission factors of major air pollutants by

vehicle classification and fuel type for that typical weekday in Macao during 2010. Average CO and
THC emission factors for gasoline powered LDPVs in Macao are significantly lower by 57% and 30%,
respectively, compared to those of gasoline LDPVs registered in Beijing, although the average driving
speed of LDPVs in Macao is lower than Beijing (e.g., ~22 km h$^{-1}$ vs. 30 km h$^{-1}$). A major reason for that
estimation is a majority of the gasoline cars are imported from Japan, where vehicle emission standards
are in general more stringent than those implemented in Mainland China (Wang et al., 2014). By
contrast, compared to gasoline taxis in Beijing, diesel engines applied in the taxi fleet in Macao led to
significantly higher NO$_X$ and PM$_{2.5}$ emission factors by 3.5 times and 17 times (Hu et al., 2012; Zhang
et al., 2014a). For heavy-duty trucks and buses, lower speed and a higher proportion of older vehicles



result in higher $NO_X$ and $PM_{2.5}$ emission factors for those heavy-duty diesel vehicles in Macao than
those in Beijing. For MCs, in particular light-duty two-stroke MCs, their fleet-average THC emission
factors are significantly higher than other vehicle technology types (Zhou et al., 2014).

4       Estimated total vehicular emissions in a typical weekday during 2010 are 17.5 tons of CO, 3.60

tons of THC, 5.04 tons of $NO_X$ and 0.28 tons of $PM_{2.5}$. As Fig. 4 illustrates, emission allocation patterns
by vehicle classification are different for various pollutant categories. Compared to well-controlled CO
and THC emission factors of LDPVs, MCs are estimated to have been responsible for 66% and 72% of
total vehicular emissions for CO and THC respectively. In particular, two-stroke MCs contribute 45% of
total THC vehicular emissions, which led Macao government to initiate a replacement of two-stroke
MCs with small-size four-stroke MCs after 2010. Further, a possible promotion of electric MCs in
Macao is also under consideration by policy-makers in Macao. For both $NO_X$ and $PM_{2.5}$, diesel-powered
passenger fleets contributed 60~65% of total vehicular emissions, including PBs, taxis and HDPVs
mainly owned by hotels and casinos. By contrast, diesel trucks contributed approximately ~10% of total
$NO_X$ and $PM_{2.5}$ emissions in Macao, substantially lower than the contribution of diesel trucks registered
in other populated cities of China (e.g., 30~35% for Beijing and Guangzhou) (Zhang et al., 2013 and
2014a). This phenomenon should be attributed to the significantly higher passenger transportation
demand than freight transportation in Macao, as tourism and entertainment industry is the pillar of the
local economy. Our results clearly suggest policy-makers in Macao should carefully focus on various
vehicle classifications when facing emission mitigation targets for various air pollutants.

20       For $CO_2$ emissions, unfavorable operating conditions like lower driving speeds and frequent use of

air-conditioning systems resulted in substantial climate and energy penalties for passenger vehicles (e.g.,
LDPVs, taxis, PBs). For example, the estimated average $CO_2$ emission factor of LDPVs is 263 g $km^{-1}$
(see Table 4), a significant increase of approximately 25% compared to on-road measurement results
under a higher average speed (205~210 g $km^{-1}$ at 30 km $h^{-1}$). This is equivalent to ~13 L per 100 km
fuel consumption, indicating a substantial increase of $CO_2$ and fuel consumption under real-world
driving conditions than those measured under the type-approval conditions applied in current regulatory
systems (e.g., both Japan and Europe). Overall, the estimated total $CO_2$ emissions from all vehicle
classifications and all road links are 1001 tons during a typical day. LDPVs, PBs and taxis are estimated
to have been responsible for 46%, 14% and 12% of total daily $CO_2$ emissions, respectively (see Fig. 4),
ranking in the top three among all classifications.





Our previous evaluation indicates estimated macro uncertainty (i.e., annual emission inventory by
using registration data) for air pollutants (e.g., CO, THC, $NO_X$ and $PM_{2.5}$) is approximately -30%/+50%
at a 95% confidence level (Zhang et al., 2014a). The skewed probability distribution is due to high
emitters of air pollutants within the fleet. The uncertainty in $CO_2$ emissions would be narrower due to
detailed localized vehicle information and fuel economy data are used in estimation, plus it is strongly
corrected by average speed. However, if the evaluation level is refined into a link-level, the uncertainty
in vehicle emissions would be greater due to traffic flows became inherently greater as the spatial
resolution was enhanced. We could address the uncertainty in link-level vehicle emissions with the
traffic big data (see the discussion in the next sub-section) available for typical roads in the future.
**3.2 Temporal and spatial variations in traffic-related emissions**
High strong correlations between temporal variations in traffic activity and emissions are clearly
observed for all air pollutants and $CO_2$ ($R^2$>0.92, see Fig. 5). For example, the 6 p.m. hour contributed
6.9% of total daily traffic activity, when hourly emissions of gaseous species (CO, THC, $NO_X$ and $CO_2$)
were responsible for 7.9%~8.7% of their daily emissions. This was because emission factors of gaseous
pollutants and $CO_2$ were increased during the rush hours due to lower driving speed. The increases were
15%~26% for their emission factors compared to the daily averages. Compared with the night time,
average gaseous emission factors of the total fleet were increased by 54%~120%. The elevation of $PM_{2.5}$
emissions in the rush hour was not as significant as gaseous species, because the traffic demand of diesel
fleets (e.g., HDPVs, taxis, PBs, trucks) was increased less relative to gasoline fleets (e.g., MCs, LDPVs)
in Macao.
Spatial distributions of vehicular emissions are associated with real-world traffic characteristics
including total traffic counts, traffic conditions and fleet composition. To sum up, 58% of $NO_X$, 52% of
$PM_{2.5}$ and 59% of $CO_2$ vehicular emissions were estimated from the road network of the MP (see Fig. 6
for $NO_X$, Fig. S6 for other pollutants and Table S2 for the summary of spatial distribution). Meanwhile,
74% of CO and 77% of THC emissions were aggregated from on-road vehicles within the MP. The
discrepancy of emission spatial allocations between CO/THC and $NO_X$/$PM_{2.5}$/$CO_2$ is primarily because
the higher fleet penetration of MCs in the MP. That is to say, relative inaccuracy associated with
emission spatial allocation by the top-down approach could be up to 20% if real-world fleet composition
information is not taken into account. By contrast, the spatial allocations of $NO_X$, $PM_{2.5}$ and $CO_2$ at three
cross-sea bridges were estimated to be higher by approximately 50~110% than CO and THC, because





the traffic volume fraction of MCs was significantly lower than in other regions, in particular compared
with the MP.
Detailed statistical profiles of spatial-related vehicular emission are summarized by length-specific
emission intensity of road groups and area-specific emission intensity of gridded cells (see Table 5 and
Table 6). Higher length-specific emission intensities of CO and THC are unexpectedly identified on
arterial roads in the MP with less traffic accounts compared with their urban freeway counterparts,
owing to higher traffic activity of MCs and more severe traffic congestion increasing all-fleet emission
factors. For $NO_X$, $PM_{2.5}$ and $CO_2$, higher length-specific emission intensities are all associated with
higher level of service for the three road classes, both in the MP and the TCC. Area-specific emission
intensities of all pollutants and $CO_2$ had decreasing trends from north to south (i.e., from the MP to the
Coloane Island), similar to the patterns of road density and traffic demand. Emission hotspots are
identified in traffic-populated cells of the MP, e.g., the region close the Ruins of St. Paul's, where daily
area-specific emission intensity was as high as 600 kg $km^{-2}$ $d^{-1}$. This level is ~4 times of that in the entire
Macao and ~40 times of the Coloane Island. Not surprisingly, significant near-field air pollution
problems in MP are caused by those extremely higher vehicular emissions which will be in addressed in
detail in the following sub-section.
It should be noted that increasingly board application of an intelligent traffic system (ITS) and
smart vehicle technologies can play a significant role in improving our understanding of dynamic traffic
flows, namely enabling the big data collection regarding total traffic volume, fleet composition and
traffic conditions (e.g., speed). For example, the traffic loop detector (TLD) and the vehicle license plate
recognition (VLPR) are both widely-used and economic ITS technologies that began in the early 2000s
in China and are integrated to provide category-informed vehicle volume, on which many cities in China
(e.g., Beijing, Guangzhou) depend to release official data including year-by-year variations in total
urban traffic demand (BJTRC, 2013; Zhang et al., 2013). The traffic loop detector is able to provide
vehicle passing speed, however, which is often criticized due to the poor representative for the entire
trips or entire traffic network. The floating car system, namely using the taxi fleet as probe vehicles
based on GPS technology, is an advanced monitoring tool for real-time traffic conditions. Taking
Beijing for example, its floating car system is capable of mapping link-based traffic conditions for the
urban area (~1000 $km^2$) every five minutes based on 66 thousand taxis and mesh urban average speed
layer down at a link level. During 2012, 24-h average speeds of the urban area of Beijing were estimated
at 23.2±2.3 km $h^{-1}$ for weekdays and 26.9±3.9 km $h^{-1}$ for weekends and holidays, respectively



(BJTRC, 2014; Zhang et al., 2014a and 2014b). Therefore, daily variations in traffic conditions could
result in a coefficient of variation (i.e., the ratio of standard deviation to mean value) of 6% for the
distance-specific $CO_2$ emission factor all year around. Most recently, the radio frequently identification
(RFID) technology has been applied in a few Chinese cities (e.g., Nanjing, the capital city of Jiangsu
province) to provide more accurate vehicle recognition with detailed specifications (e.g., category, fuel
type, emission standard, model year, and vehicle size) than the TLD and the VLPR. The RFID data in
Nanjing are further connected with a smartphone application, based on which more capabilities like
environmentally-constrained traffic management (e.g., low emission zone, congestion fee program)
could be developed in the future. From the perspective of vehicles, for instance, more real vehicle data
can be accessed through the on-board diagnostic (OBD) decoders. The second-by-second data of driving
conditions (e.g., speed, acceleration) are able to be combined with operating mode-based (e.g.,
VSP-informed) emission model to provide finer emission estimations. While foregoing advanced traffic
data collection methods are not available in Macao, the framework of this study is technically feasible to
large cities in China when the traffic big data are adequately available.
**3.3 Simulated concentrations of primary traffic-related pollutants in Macao**
Fig. 7 presents a spatial map of average concentrations of primary vehicle-contributed $NO_2$ (see
CO and $PM_{2.5}$ in Fig. S7), which shows the simulated results of all receptors (i.e., central points of cells)
with the AERMOD model. The spatial variations in simulated concentrations highly resemble the
patterns of area-specific emission intensity for vehicular pollutants. For example, average concentrations
contributed by local vehicular emissions in Macao were $87.7 \pm 89.4$ μg m$^{-3}$ of CO, $22.2 \pm 17.1$ μg m$^{-3}$ of
$NO_2$ and $1.30 \pm 0.91$ μg m$^{-3}$ of $PM_{2.5}$, respectively (see Table 6). Highest receptor concentrations of CO,
$NO_X$ and $PM_{2.5}$ are 424, 84.9 and 4.42 μg m$^{-3}$, respectively, all occurring at traffic-populated cells in the
MP.
We further compared modeled concentrations of primary pollutants from local vehicles and
official air quality data. Traffic contributions at the monitoring sites are approximated by simulated
results for their closest receptors as to estimate monthly-average source proportions of on-road vehicles
in Macao. Therefore, source proportions vary from pollutant categories and locations during the time
framework of this study. For example, estimated proportions of vehicular CO emissions are ~25-30% in
the MP and ~15% in the Taipa Island, indicating lower impacts compared to regional contributions. For
$NO_2$, significantly high local contributions are identified. For example, for two monitoring sites in





residential areas, 50-55% of monitored $NO_2$ concentrations are caused by local vehicles. The proportion for the site located in traffic-populated area is estimated over 80%. The contribution proportion of on-road vehicles is estimated predominantly in the traffic-dense area. With regard to $PM_{2.5}$, estimated proportions of primary vehicular $PM_{2.5}$ emissions are approximately 3% in residential areas and 5% in traffic-populated area. It should be acknowledged that the atmospheric secondary $PM_{2.5}$ considerably contributed by vehicle emissions is not considered in this study, which need to be applied with a very detailed regional emission inventory including all anthropogenic emission sources and complex air quality models with sophisticated source apportionment functions. This is beyond the scope of this paper.

Hence, we compared daily $NO_2$ concentrations during 19 weekdays of November 2010 between simulated results and observed concentrations for one monitoring site in the MP (see Fig. 8), because another air monitoring site in the MP is very close to the city boundary (i.e., less than 300m) and greatly influenced by vehicle emissions in Zhuhai (i.e., the adjacent city to Macao). We observe a reasonable correlation (Pearson's R=0.67) between simulated results and observed concentration for this site. The average discrepancy between simulated and observed concentrations is $15 \pm 22\%$, which may probably be attributed to the following aspects but not limited to: (1) uncertainty in estimating background level and regional transport by using CMAQ since Macao only covers two CMAQ cells; (2) the approximation of the receptor of AERMOD and the air quality monitoring sites; (3) the strong street-canyon effects in the building-dense MP which are not sophisticatedly addressed by the AERMOD. For example, Sheng and Tang (2011) coupled the OSPM model and detailed building-based geography layer and derived hotspots of traffic-related $NO_2$ concentrations that were simulated higher than 62 ppb (i.e., 127 µg m$^{-3}$) in 2004. Although their emission factors estimated with the MOBILE model (Tang et al., 2006) were lower than ours based on local PEMS measurement, their higher results could be attributed to the significant street canyon effect and higher spatial resolution (i.e., 319 receptors km$^{-2}$) compared with our study. Furthermore, specifically for hotspots, advanced computational fluid dynamics (CFD)-based micro-scale air quality model coupled with sophisticated gaseous chemical mechanisms and aerosol dynamics are suggested to quantitatively assess potential impacts and mitigation strategies from perspectives of traffic flows, weather conditions and architecture layout. Given the severe traffic congestion in Macao, which is an unfavorable condition for some advanced $deNO_X$ after-treatment devices (e.g., selective catalyst reduction for diesel commercial vehicles), other effective mitigation alternatives should be carefully considered by local policy-makers such as a



substantial penetration of alternative fuel and advanced powertrain systems to the public fleets in Macao
(e.g., dedicated natural gas buses, hybrid electric taxis and battery electric taxis/buses) (Zhang et al.,
2014d; Wang et al., 2015; Wu et al., 2015).
**4. Conclusions**
High-resolution vehicle emission inventory is an irreplaceable assessment tool to achieve the fine
air quality administration, in particular for traffic-populated East Asian cities where traffic management
is an essential approach to reduce emissions. Due to the difficulties in obtaining link-level traffic flow
data and localized emission measurement profiles, such a dedicated environmental tool has not been
developed at the link-level which covers a whole city and all vehicle categories. This study selected the
entire area of Macao, the most populated city in this world, to demonstrate a high-resolution simulation
of vehicular pollution by coupling detailed local data collected and inter-disciplinary models (e.g., traffic
demand).
Our traffic flow investigation and simulation results showed that total daily traffic activity during a
typical weekday of 2010 was estimated at 4.06 million veh km d$^{-1}$. Passenger trips using MCs, LDPVs,
taxis and buses were responsible for a dominant part of travel demand in Macao, accompanied by a
significantly less traffic fraction of on-road freight transportation (e.g., trucks) than other cities in
Mainland China. Spatial heterogeneity of traffic flow characteristics has been discerned between the MP
and the remaining parts (i.e., the TCC) of Macao. For example, the MP contributed over 80% of total
traffic accounts in Macao during a weekday of 2010 and MCs were more prevalent in this more
populated peninsula compared to the TCC. Tremendous travel demand created during rush hours
resulted in significant traffic congestion, indicated by an average speed lower than 15 km h$^{-1}$ for arterial
and residential roads in the MP.
Based on a localized vehicle emission model (e.g., the EMBEV-Macao) and high-resolution traffic
profiles regarding traffic volume, average speed and fleet composition, this study established a
link-based vehicle emission inventory with high resolution meshed in a temporal and spatial framework
(e.g., hourly and link-level). We estimated that total daily vehicle emissions in Macao were 17.5 tons of
CO, 3.60 tons of THC, 5.04 tons of NO$_X$ and 0.28 tons of PM$_{2.5}$ during a typical weekday of 2010. MCs
are the major contributor to CO and THC emissions due to their higher emission factors than LDPVs.
Diesel-powered passenger fleets like buses and taxis contributed 60~65% of total vehicular emissions of
NO$_X$ and PM$_{2.5}$. With a special focus on the MP region, where traffic density and congestion are more



significant, area-specific emission intensity can be higher than the average of the entire Macao area by 135% for CO, 145% for THC, 85% for $NO_X$, 65% for $PM_{2.5}$ and 90% for $CO_2$. The geographic discrepancy of spatial allocation between THC and $PM_{2.5}$ emissions can be attributed to the spatially heterogeneous vehicle-use intensity between MCs and diesel fleets (e.g., higher use intensity of MCs in the MP); and this trait could not be identified by using the traditional emission inventory tool. From the perspective of temporal variations, hourly emissions of CO, THC, $NO_X$ and $CO_2$ during the evening traffic peak could be responsible for 7.9%~8.7% of total daily emissions, when their emission factors were increased by 15%~26% compared to the daily averages due to the traffic congestion.

We further employed the AERMOD model to quantify average concentrations of CO, $NO_2$ and $PM_{2.5}$ contributed by primary vehicle emissions in Macao. Our simulation indicated receptor-averaged concentrations from primary vehicle emissions were $87.7 \pm 89.4$ μg m$^{-3}$ of CO, $22.2 \pm 17.1$ μg m$^{-3}$ of $NO_2$ and $1.30 \pm 0.91$ μg m$^{-3}$ of $PM_{2.5}$, respectively, during the weekdays of November, 2010. The highest receptor concentrations of CO, $NO_X$ and $PM_{2.5}$ were 424 μg m$^{-3}$, 84.9 μg m$^{-3}$ and 4.42 μg m$^{-3}$, respectively, all occurring at traffic-populated cells in the MP. On-road vehicles are a dominant source of ambient $NO_2$ in traffic-populated areas as indicated by the good agreement between AERMOD-simulated data and observed results. This paper can provide a useful case study and a solid framework for developing high-resolution environmental assessment tools for other vehicle-populated cities in the world. We also highlighted the importance of real traffic data using ITS techniques and the traffic big data approaches to future high-resolution simulation for larger cities in the East Asia and all over the world.

*Acknowledgments.* This work was sponsored by the National High Technology Research and Development Program (863) of China (No. 2013AA065303), the National Natural Science Foundation of China (No. 51322804 and No. 51378285), and the Program for New Century Excellent Talents in University (NCET-13-0332) . We thank Mr. Jiandong Wang and Miss Xiao Fu of Tsinghua University for their great help in running the CMAQ Model. The contents of this paper are solely the responsibility of the authors and do not necessarily represent official views of the sponsors.



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

| Region | The Macao Peninsula | | | The Taipa-CoTai-Coloane Region | | |
|---|---|---|---|---|---|---|
| Road classes | Freeway | Arterial | Residential | Freeway | Arterial | Residential |
| 0 | 0.021 | 0.017 | 0.021 | 0.021 | 0.017 | 0.022 |
| 1 | 0.013 | 0.014 | 0.013 | 0.013 | 0.014 | 0.013 |
| 2 | 0.011 | 0.009 | 0.011 | 0.011 | 0.010 | 0.011 |
| 3 | 0.009 | 0.007 | 0.009 | 0.009 | 0.007 | 0.009 |
| 4 | 0.008 | 0.007 | 0.008 | 0.008 | 0.007 | 0.008 |
| 5 | 0.008 | 0.008 | 0.008 | 0.008 | 0.008 | 0.008 |
| 6 | 0.021 | 0.024 | 0.020 | 0.021 | 0.024 | 0.021 |
| 7 | 0.029 | 0.051 | 0.029 | 0.029 | 0.022 | 0.030 |
| 8 | 0.051 | 0.057 | 0.059 | 0.048 | 0.053 | 0.061 |
| 9 | 0.048 | 0.054 | 0.048 | 0.042 | 0.052 | 0.051 |
| 10 | 0.044 | 0.049 | 0.050 | 0.046 | 0.055 | 0.049 |
| 11 | 0.055 | 0.050 | 0.049 | 0.056 | 0.056 | 0.048 |
| Hour 12 | 0.051 | 0.056 | 0.055 | 0.051 | 0.056 | 0.058 |
| 13 | 0.059 | 0.062 | 0.061 | 0.062 | 0.064 | 0.062 |
| 14 | 0.060 | 0.066 | 0.064 | 0.070 | 0.073 | 0.059 |
| 15 | 0.064 | 0.061 | 0.059 | 0.068 | 0.072 | 0.065 |
| 16 | 0.066 | 0.061 | 0.060 | 0.071 | 0.070 | 0.046 |
| 17 | 0.066 | 0.066 | 0.059 | 0.065 | 0.069 | 0.069 |
| 18 | 0.071 | 0.066 | 0.076 | 0.062 | 0.060 | 0.070 |
| 19 | 0.061 | 0.057 | 0.062 | 0.054 | 0.051 | 0.075 |
| 20 | 0.049 | 0.045 | 0.052 | 0.049 | 0.046 | 0.045 |
| 21 | 0.048 | 0.041 | 0.052 | 0.048 | 0.042 | 0.050 |
| 22 | 0.047 | 0.039 | 0.042 | 0.047 | 0.039 | 0.039 |
| 23 | 0.042 | 0.033 | 0.032 | 0.042 | 0.034 | 0.033 |



1 **Table 2.** Summary of age allocation for on-road fleets by vehicle classification in Macao

| Vehicle classification | | LDPV | | MC | | Taxi | PB | | MDPV | | HDPV | LDT | | HDT |
|---|---|---|---|---|---|---|---|---|---|---|---|---|---|---|
| Sub-classification | | G [a] | D [b] | Heavy [c] | Light [c] | D | Medium [d] | Heavy [d] | G | D | D | G | D | D |
| Ratio | | 0.99 | 0.01 | 0.68 | 0.32 | 1.00 | 0.33 | 0.67 | 0.53 | 0.47 | 1.00 | 0.25 | 0.75 | 1.00 |
| Vehicle age | 1 | 0.12 | 0.12 | 0.18 | 0.09 | 0.14 | 0.00 | 0.08 | 0.20 | 0.16 | 0.20 | 0.12 | 0.08 | 0.02 |
| | 2 | 0.10 | 0.17 | 0.15 | 0.08 | 0.13 | 0.00 | 0.08 | 0.17 | 0.17 | 0.06 | 0.17 | 0.18 | 0.15 |
| | 3 | 0.10 | 0.08 | 0.19 | 0.09 | 0.04 | 0.00 | 0.08 | 0.07 | 0.12 | 0.09 | 0.11 | 0.10 | 0.11 |
| | 4 | 0.10 | 0.11 | 0.14 | 0.07 | 0.06 | 0.00 | 0.18 | 0.06 | 0.02 | 0.10 | 0.03 | 0.09 | 0.04 |
| | 5 | 0.09 | 0.03 | 0.08 | 0.04 | 0.06 | 0.17 | 0.16 | 0.05 | 0.09 | 0.09 | 0.03 | 0.05 | 0.03 |
| | 6 | 0.06 | 0.05 | 0.05 | 0.07 | 0.02 | 0.12 | 0.14 | 0.05 | 0.03 | 0.09 | 0.09 | 0.04 | 0.01 |
| | 7 | 0.05 | 0.01 | 0.04 | 0.04 | 0.11 | 0.25 | 0.15 | 0.06 | 0.01 | 0.03 | 0.00 | 0.02 | 0.01 |
| | 8 | 0.05 | 0.02 | 0.04 | 0.07 | 0.16 | 0.05 | 0.05 | 0.08 | 0.01 | 0.05 | 0.05 | 0.02 | 0.00 |
| | 9 | 0.04 | 0.03 | 0.02 | 0.08 | 0.24 | 0.00 | 0.00 | 0.04 | 0.01 | 0.05 | 0.02 | 0.02 | 0.01 |
| | 10 | 0.04 | 0.06 | 0.01 | 0.13 | 0.01 | 0.07 | 0.00 | 0.06 | 0.02 | 0.04 | 0.01 | 0.03 | 0.02 |
| | 11 | 0.05 | 0.06 | 0.03 | 0.14 | 0.03 | 0.17 | 0.01 | 0.02 | 0.01 | 0.10 | 0.02 | 0.04 | 0.01 |
| | 12 | 0.05 | 0.04 | 0.02 | 0.06 | 0.00 | 0.00 | 0.03 | 0.03 | 0.01 | 0.04 | 0.01 | 0.04 | 0.02 |
| | 13 | 0.03 | 0.06 | 0.00 | 0.01 | 0.00 | 0.03 | 0.00 | 0.02 | 0.03 | 0.00 | 0.02 | 0.02 | 0.01 |
| | 14 | 0.04 | 0.05 | 0.01 | 0.01 | 0.00 | 0.10 | 0.00 | 0.02 | 0.03 | 0.00 | 0.06 | 0.04 | 0.04 |
| | 15 | 0.03 | 0.05 | 0.01 | 0.01 | 0.00 | 0.05 | 0.00 | 0.04 | 0.05 | 0.00 | 0.06 | 0.04 | 0.11 |
| | 16 | 0.02 | 0.02 | 0.01 | 0.00 | 0.00 | 0.00 | 0.00 | 0.03 | 0.04 | 0.03 | 0.04 | 0.06 | 0.16 |
| | 17 | 0.01 | 0.03 | 0.00 | 0.01 | 0.00 | 0.00 | 0.00 | 0.00 | 0.03 | 0.00 | 0.05 | 0.04 | 0.06 |
| | 18 | 0.01 | 0.01 | 0.00 | 0.00 | 0.00 | 0.00 | 0.00 | 0.00 | 0.06 | 0.00 | 0.05 | 0.03 | 0.03 |
| | 19 | 0.00 | 0.00 | 0.00 | 0.00 | 0.00 | 0.00 | 0.04 | 0.00 | 0.02 | 0.00 | 0.02 | 0.02 | 0.07 |
| | 20 | 0.00 | 0.01 | 0.00 | 0.00 | 0.00 | 0.00 | 0.00 | 0.00 | 0.07 | 0.02 | 0.03 | 0.05 | 0.08 |
| Fleet-average vehicle age | | 6.7 | 7.3 | 4.4 | 7.2 | 5.8 | 8.6 | 5.5 | 5.7 | 7.9 | 6.0 | 8.1 | 8.1 | 11.4 |

2 Note: [a] gasoline; [b] diesel; [c] breaking point of engine displacement 50 ml; [d] breaking point of engine displacement at 5.0 L.





1 **Table 3.** Spatially-explicit estimation of traffic counts in Macao

| Region | Daily traffic counts by road class ($10^5$ veh) | | | Hour-based density of traffic volume ($10^4$ veh h$^{-1}$ km$^{-2}$) | |
|---|---|---|---|---|---|
| | Freeway | Arterial | Residential | Daily average | Evening rush hour (6 p.m.) |
| Macao Peninsula | 15.2 | 70.8 | 138.4 | 10.0 | 17.3 |
| Saint Antony Parish | 2.8 | 20.5 | 35.0 | 25.3 | 44.3 |
| Taipa-Cotai-Coloane | 6.9 | 13.9 | 28.8 | 1.0 | 1.5 |
| Taipa | 2.2 | 12.5 | 17.8 | 2.0 | 3.1 |
| Cotai | 3.6 | 1.4 | 7.1 | 0.8 | 1.3 |
| Coloane | 1.1 | | 3.9 | 0.3 | 0.5 |
| Total | 22.2 | 84.7 | 170.2 | 3.8 | 6.5 |

3 **Table 4.** Estimated fleet-average emission factors under real-world driving conditions

| Vehicle classification | Fleet-average emission factors (g km$^{-1}$) | | | | |
|---|---|---|---|---|---|
| | CO | THC | NO$_X$ | PM$_{2.5}$ | CO$_2$ |
| LDPV-Gasoline | 1.74 | 0.34 | 0.28 | 0.006 | 263 |
| MDPV-Gasoline | 14.3 | 1.80 | 1.18 | 0.030 | 379 |
| MDPV-Diesel | 1.60 | 0.27 | 1.44 | 0.26 | 307 |
| HDPV-Diesel | 4.76 | 0.25 | 10.9 | 0.48 | 914 |
| LDT-Gasoline | 8.38 | 2.30 | 1.31 | 0.014 | 250 |
| LDT-Diesel | 1.69 | 0.65 | 4.03 | 0.35 | 485 |
| HDT-Diesel | 7.40 | 0.94 | 12.3 | 0.95 | 1010 |
| Taxi | 0.47 | 0.06 | 0.86 | 0.11 | 192 |
| MC-Light | 7.95 | 4.07 | 0.26 | 0.030 | 39 |
| MC-Heavy | 10.2 | 1.18 | 0.38 | 0.012 | 86 |
| PB-Medium | 2.45 | 1.09 | 6.50 | 0.32 | 555 |
| PB-Heavy | 6.05 | 0.35 | 15.8 | 0.57 | 1215 |





**Table 5.** Length-specific emission intensity of total vehicular emissions during a typical
weekday of 2010

| Region | Road class | Length-specific emission intensity (kg km$^{-1}$ d$^{-1}$) | | | | |
|---|---|---|---|---|---|---|
| | | CO | THC | NO$_X$ | PM$_{2.5}$ | CO$_2$ |
| Macao Peninsula | Freeway | 147 | 28 | 43 | 2.6 | 9046 |
| | Arterial | 196 | 42 | 39 | 1.9 | 7819 |
| | Residential | 80 | 17 | 18 | 0.9 | 3741 |
| Taipa-Cotai-Coloane | Freeway | 78 | 13 | 43 | 2.9 | 7412 |
| | Arterial | 58 | 10 | 36 | 2.3 | 5948 |
| | Residential | 25 | 5 | 6 | 0.4 | 1909 |
| Cross-sea bridges | Freeways | 130 | 24 | 61 | 4.0 | 10813 |
| Total | Freeway | 117 | 22 | 49 | 3.1 | 9069 |
| | Arterial | 124 | 25 | 38 | 2.1 | 6846 |
| | Residential | 60 | 13 | 14 | 0.7 | 3081 |

**Table 6.** Area-specific emission intensity of total vehicular emissions during a typical
weekday of 2010

| Region / Parish | Area-specific emission intensity (kg km$^{-2}$ d$^{-1}$) | | | | |
|---|---|---|---|---|---|
| | CO | THC | NO$_X$ | PM$_{2.5}$ | CO$_2$ |
| Macao Peninsula | 1387 | 297 | 312 | 15.5 | 63695 |
| St. Lazarus Parish | 3152 | 682 | 696 | 33.7 | 139276 |
| St. Lawrence Parish | 1421 | 303 | 305 | 15.4 | 61268 |
| Our Lady Fatima Parish | 1258 | 271 | 274 | 13.7 | 57362 |
| St. Anthony Parish | 2520 | 547 | 557 | 26.4 | 117967 |
| Cathedral Parish | 799 | 166 | 199 | 10.4 | 38585 |
| Taipa | 301 | 53 | 151 | 9.50 | 27977 |
| CoTai Reclamation Area | 163 | 28 | 71 | 4.67 | 14055 |
| Coloane | 51 | 11 | 15 | 0.88 | 4440 |
| Total land area of Macao | 590 | 121 | 169 | 9.42 | 33645 |





1  **Table 7.** Simulated average contributions contributed by primarily vehicular emissions in Macao, weekdays during November 2010

| Region / Parish | Simulated concentrations of primary vehicular emissions (μg m$^{-3}$) | | | | | | | | |
|---|---|---|---|---|---|---|---|---|---|
| | CO | | | NO$_2$ | | | PM$_{2.5}$ | | |
| | Mean | Min | Max | Mean | Min | Max | Mean | Min | Max |
| Macao Peninsula | 205 | 59.5 | 424 | 42.3 | 14.3 | 84.9 | 2.03 | 0.67 | 3.89 |
| St. Lazarus Parish | 340 | 277 | 424 | 68.4 | 57.1 | 84.9 | 3.14 | 2.59 | 3.89 |
| St. Lawrence Parish | 186 | 143 | 272 | 37.1 | 27.9 | 52.6 | 1.72 | 1.32 | 2.47 |
| Our Lady Fatima Parish | 176 | 79.9 | 317 | 35.4 | 15.2 | 58.5 | 1.64 | 0.67 | 3.21 |
| St. Anthony Parish | 305 | 2306 | 372 | 60.5 | 44.7 | 74.5 | 2.85 | 2.17 | 3.39 |
| Cathedral Parish | 171 | 59.5 | 380 | 34.1 | 14.3 | 78.7 | 2.03 | 1.00 | 3.14 |
| Taipa | 44.1 | 13.2 | 109 | 16.7 | 4.32 | 40.9 | 1.65 | 0.61 | 2.46 |
| CoTai Reclamation Area | 38.9 | 11.7 | 66.1 | 14.8 | 4.10 | 27.6 | 1.08 | 0.27 | 2.39 |
| Coloane | 17.7 | 7.34 | 56.6 | 5.72 | 2.06 | 17.6 | 0.29 | 0.12 | 0.63 |
| Total land area of Macao | 87.7 | 7.34 | 424 | 22.2 | 2.06 | 84.9 | 1.30 | 0.12 | 3.89 |

2  Note: Simulated results for November 6-8 are not accounted in this table due to the impact of rainfall. Mean, minimum and maximum
3  values are for simulated average concentrations of each receptors in each region/parish during the study period.




1 **Figures**

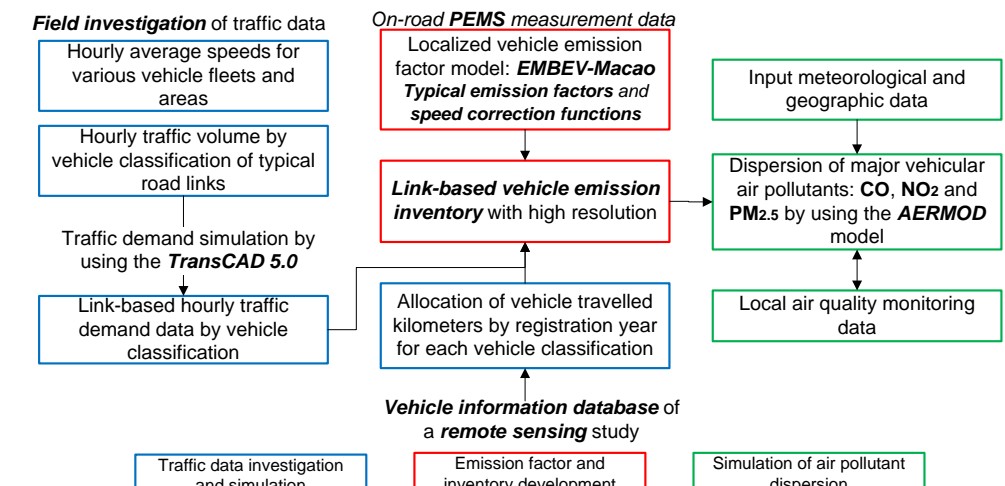

4  Fig. 1. Framework of high-resolution simulation for vehicle emissions and concentrations

5  of vehicular pollutants.



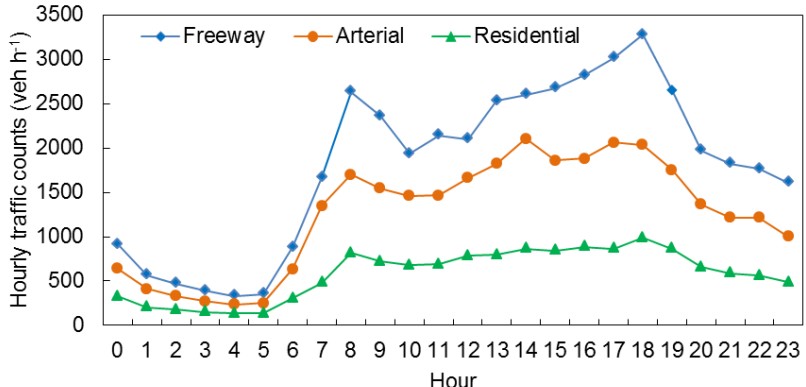

2    Fig. 2. Average hourly traffic accounts of observed links by road class during weekdays,
3    2010.
4





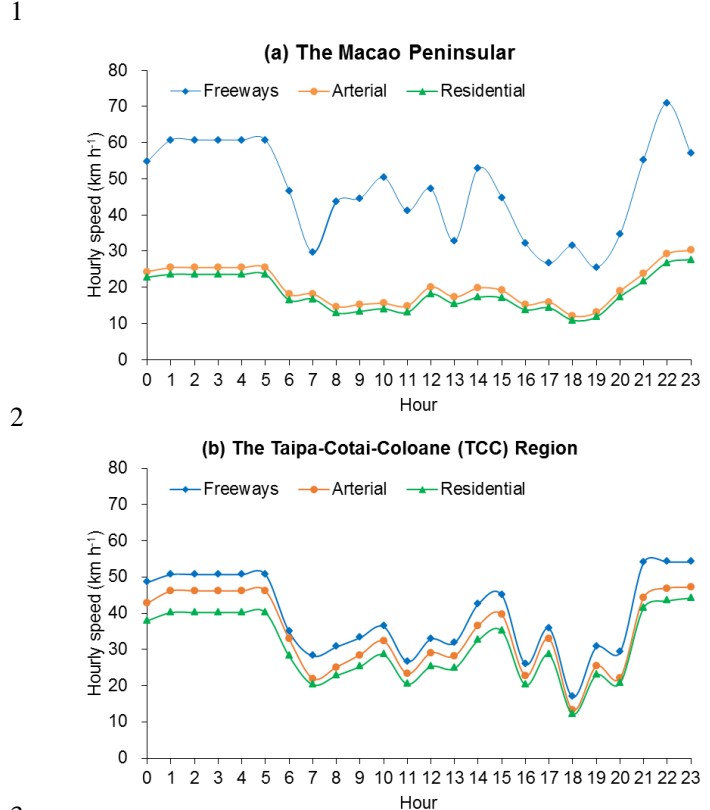

4
5    Fig. 3. Variations in aggregated hourly speeds by road class and region for LDPVs during
6    weekdays, 2010.







3
4    Fig. 4. Allocations of total vehicular emissions by vehicle classification
5



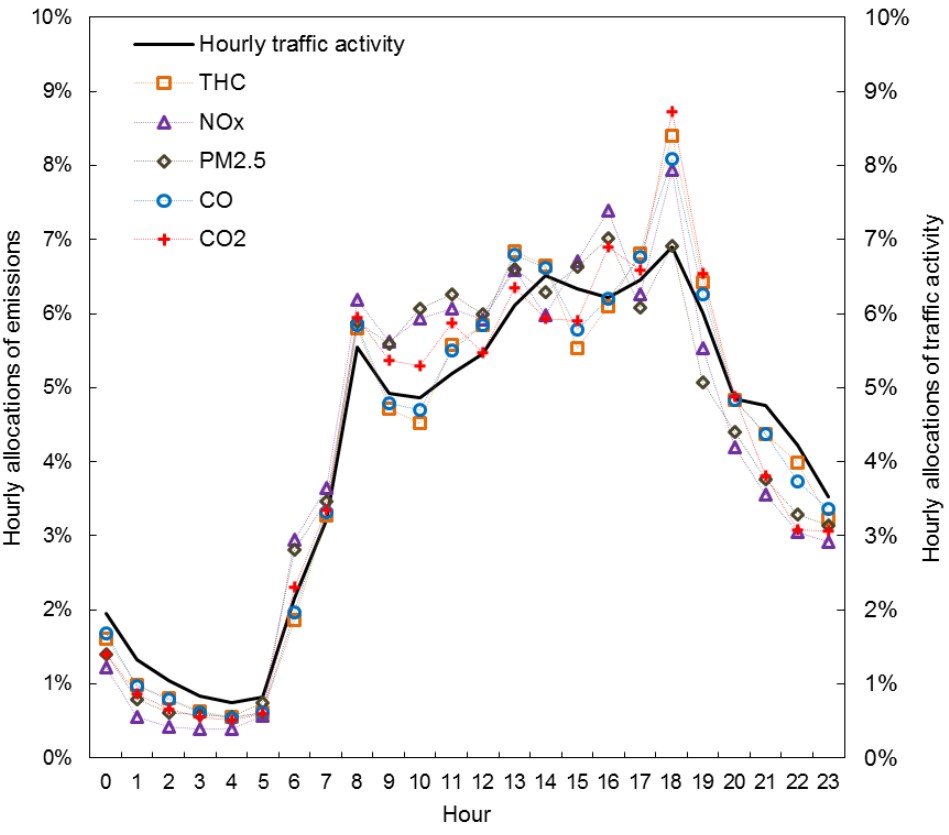

2    Fig. 5. Hourly allocations of vehicular emissions and traffic activity in Macao during
3    weekdays, 2010.



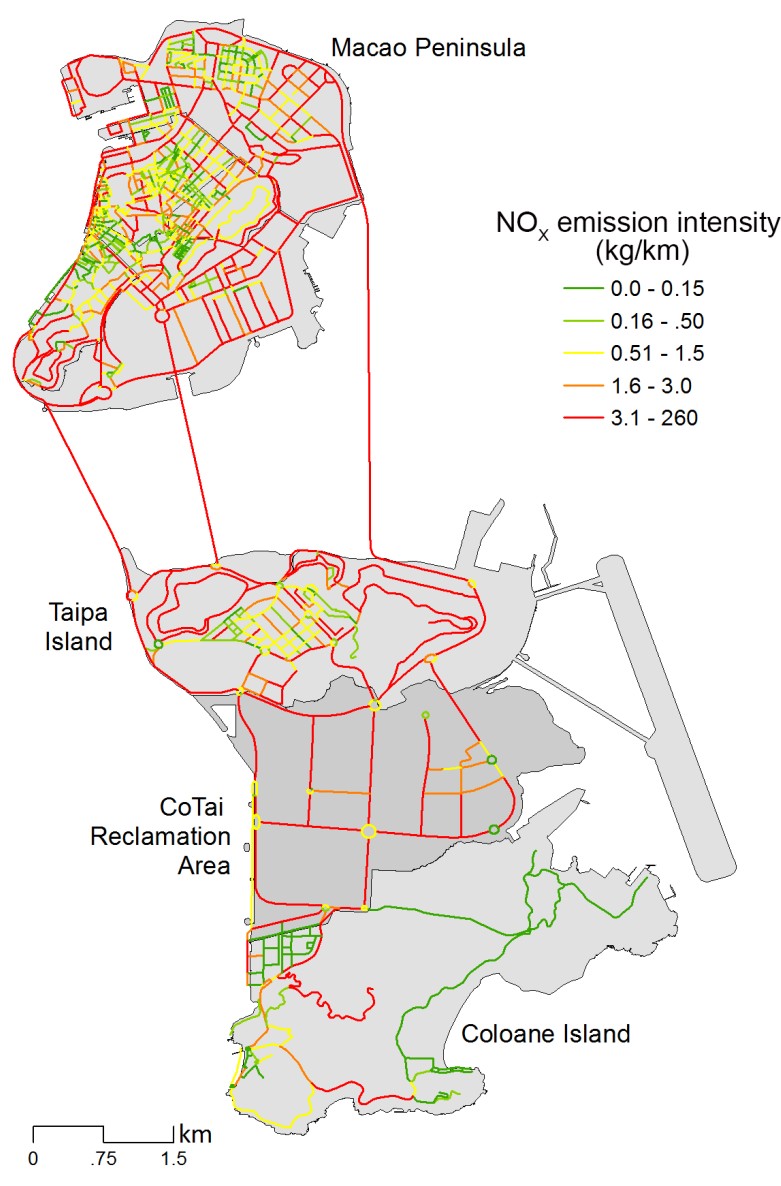

2      Fig. 6. The spatial distribution of NO$_X$ emission intensity for on-road vehicles in Macao

3      during a typical weekday of 2010





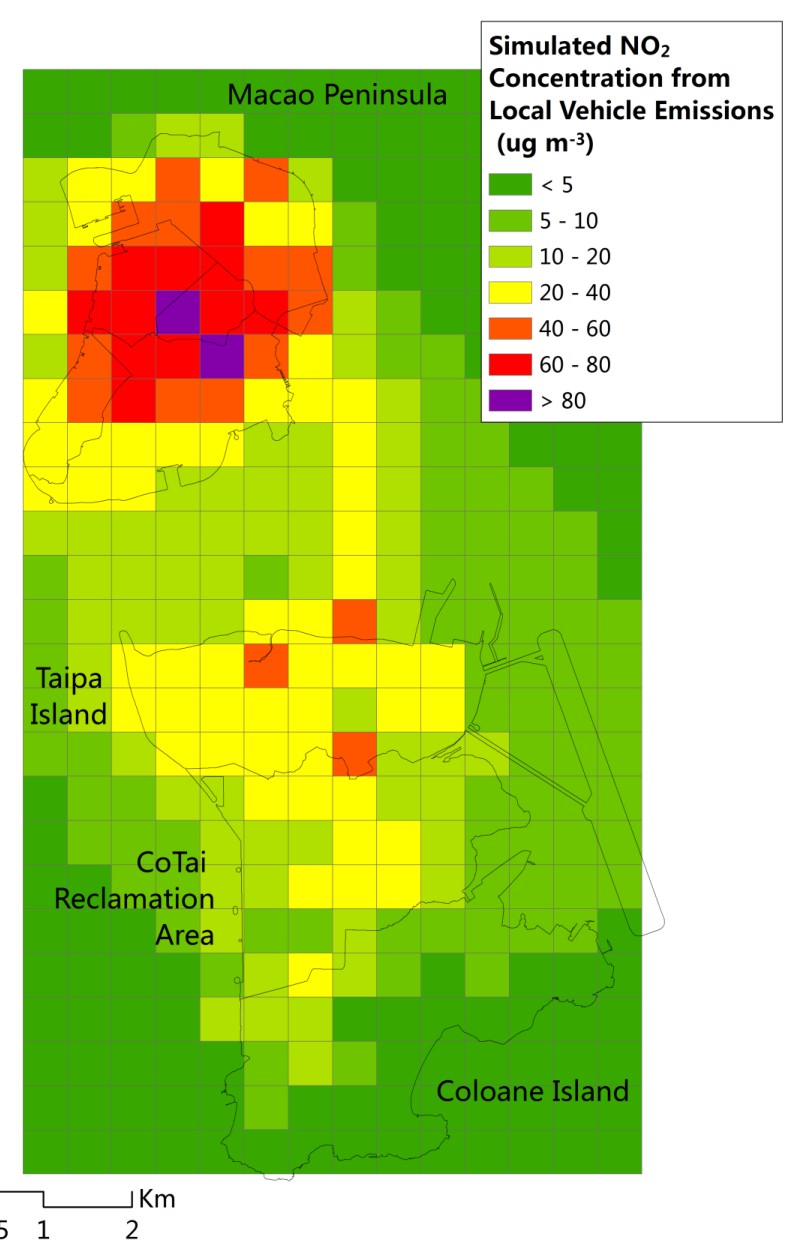

2    Fig. 7. Simulated vehicle-contributed concentration of $NO_2$ in Macao during weekdays of

3    November, 2010





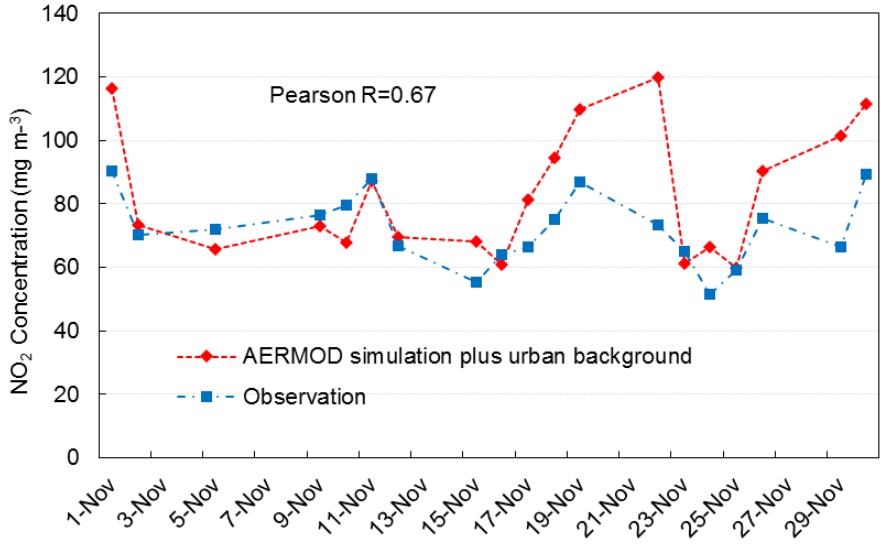

Fig. 8. Comparison of AERMOD simulated and observed daily NO$_2$ concentrations for
the traffic-populated site, 19 weekdays during November, 2010
Note: Simulated NO$_2$ concentrations during November 6-8 are significantly higher than
observed data probably due to the effect of rainfall, which are not included in this study.
