# Peer review of "High-resolution simulation of link-level vehicle emissions and"

_Atmospheric Chemistry and Physics, 2016_

## Referee Comment (RC1) · Anonymous Referee #1 · 23 Mar 2016

Overall Comments: My overall comments are favorable. I thought the paper presented interesting results of the use of a vehicle emission model to infer the dominant sources of vehicle pollution in Macao, as well as the potential use of traffic information linked to an emission model and air dispersion model to inform environmental and transportation policy. My major comment on the content of the paper, is that the authors need to provide more information into how the emissions and air quality concentrations were estimated are estimated. For example, the paper is unclear on how the vehicle split by link is estimated, how the speeds are estimated and applied in the vehicle emission model, and how the 'fleet average' emission rates in Table 4 are estimated. Also the

discussion on the air dispersion modeling is very limited.

I would also strongly recommend the authors change the wording on page 2 (line 25) and pae 17 (line 6) from 'irreplaceable' tool, to 'can be to a valuable tool'. I do not think the paper showed that the high-resolution traffic tool is an irreplaceable assessment tool. The paper did show that the results from the tool, appear to compare reasonably well with at least one air quality monitor, and it discusses ways, in which it could be used to inform air quality and policy decisions in the future. I have other minor comments that I would like to see the authors address, to improve clarity of the report, and improve the communication. Page 4. Line 4, huge a large transportation demand. Page 6. Line 21. What is the MC fraction on the Macao Peninsula? Page 7, Line 24. You should mention the variability in the speed trends across roadway links which could be due to the limited data based on chase-car study. Figure 3 appears to have significant variability from hour to hour, that I would think would be smoother if it had a larger sample size across more sample days, more links, and more vehicles. This should at least be discussed, especially if confidence intervals of the mean speed are not presented (which I think would show that many of the hourly mean speeds are not significantly different than one another). Page 11. How do you obtain estimates of vehicle classifications by link? This is not clear to me from reading the first paragraph on page 11. Page 14. Line 8-9. I think you mean higher emission rates, for lower level of service? Page 14. Line 17. 'broad' instead of 'board' Page 14. Line 25-26. Rephrase this sentence. 'poor representativeness. . ..' Page 15. Line 1-3. How does the daily variations in speeds, results in the variation in CO2 emission factors? Is that from analysis done from the Beijing study? Or is that variation in link speeds applied to your emission model for Macao? Please be clear. Page 16. Line 5-9. Rewrite sentence, and improve grammar. Page 16. Line 29. Start new paragraph. Page 17. Line 6. Suggest 'can be a valuable assessment tool' (not ' irreplaceable') Page 17. Line 17. Replace 'significantly less traffic' with 'smaller' Page 17. Line 30. The Taxis are diesel powered? This should be clarified in the main text, as well as in Table 4. RE: Table 4. I am surprised that the diesel Taxis' have lower NOx g/km, than the MDPV gasoline vehicles? Are these

emission rates based on PEMS data or emission standards? If some of these emission rates are based on certification vehicle standards, than the paper should mention the uncertainty of using vehicle emission standards (particularly EURO diesel standards) to represent real-world emission rates. Also, similarly, why is the MDPV diesel in the same range as the MDPV gasoline vehicles? In general, more information is needed on the derivation of the fleet-average emission factors in Table 4.
* * *

---

## Referee Comment (RC2) · Anonymous Referee #2 · 18 Apr 2016

This study develops a high-resolution motor vehicle emissions inventory for a city in China, models the inventory, and evaluates the modeling results against ambient monitoring data. Main findings of this paper are that it is important to capture the spatial heterogeneity of the vehicle fleet mix across the urban domain, and to use local information on emission factors. Overall, the authors present a novel approach to mapping vehicle emissions, especially in cities where traffic activity and emission factor data are not as readily available. This is a major accomplishment and worth replicating in other cities. In regards, to the second major aspect of this study, the air quality modeling, I have some concerns that I believe need to be addressed more fully in revision.

[Figure]

My comments mostly refer to the treatment of atmospheric chemistry in the dispersion model. With major revision, I do believe it is possible for this manuscript to be considered for publication in Atmospheric Chemistry & Physics.

General Comments

(1) My concerns with respect to the air quality modeling are with the treatment of chemistry, and how background levels are estimated. Unless the following concerns can be addressed, I believe that statements that quantify the fractional contribution of motor vehicle emissions to ambient concentrations observed should be removed (bottom of page 15 and top of page 16), and commentary restricted to qualitative statements.

(i) More detail is needed on how a dispersion model like AERMOD accounts for chemistry, especially for $NO_2$. Given that authors present high-resolution air quality maps, it seems important to capture spatial gradients that may arise due to interactions between ozone and fresh NO emissions; i.e., ozone tends to be suppressed near highways, and $NO_2$ elevated (Murphy et al., 2007). On page 10, Lines 15-17, the authors mention using ozone data to account for the oxidation of NO to $NO_2$, but do not describe how. How many monitoring sites are used in this calculation? Where are they located, and what is their proximity to roadways? What is the timescale of the NO -> $NO_2$ conversion employed in AERMOD, and how was this estimated from observations?

Murphy, J. G., Day, D. A., Cleary, P. A., Wooldridge, P. J., Millet, D. B., Goldstein, A. H., and Cohen, R. C.: The weekend effect within and downwind of Sacramento – Part 1: Observations of ozone, nitrogen oxides, and VOC reactivity, Atmos. Chem. Phys., 7, 5327-5339, doi:10.5194/acp-7-5327-2007, 2007.

(ii) It is also not clear how AERMOD treats the loss of $NO_2$ to PAN and $HNO_3$. In an urban mass, these products of $NO_2$ can comprise up to half of daytime $NO_y$ (= $NO_x$ + PAN + $HNO_3$, see Pollack et al., 2012). If the authors' only account for the production of $NO_2$ from fresh NO emissions, without accounting for the loss of $NO_2$ from daytime chemistry, then the $NO_2$ concentrations simulated from local vehicle emissions shown

in Figure 7 could be overestimated. Consequently, the authors may overestimate the motor vehicle contribution to ambient NO2. The importance of the loss term will depend on the photochemical age of the air mass that reaches the monitoring site, which could vary by time of day, wind speed/direction, and synoptic events. A robust estimation of the local vehicle contribution to ambient NO2 (as shown in Figure 8) should take into account both the production and loss of NO2.

Pollack, I. B., et al. (2012), Airborne and ground-based observations of a weekend effect in ozone, precursors, and oxidation products in the California South Coast Air Basin, J. Geophys. Res., 117, D00V05, doi:10.1029/2011JD016772.

(iii) As I understand, the NO2 observations shown in Figure 8, are daily concentrations from a single monitoring site (Page 16, Lines 10-13). Is this the average of 24-hours of data? The use of daily averages could be influenced by nighttime chemistry, which presumably would not be taken into account with AERMOD. To avoid these complications, it is better to restrict the model comparison to daytime values only.

(iv) I found the description of the CMAQ model (on Page 10, Lines 20-26) used to estimate regional background and cross-boundary transport lacking. This is important since background levels (Page 10: 304 ug/m3, 27 ug/m3, and 23 ug/m3 of CO, NO2, and PM2.5, respectively) are as big or much larger than the motor vehicle contribution to these pollutants (Page 15: 88 ug/m3, 22 ug/m3, and 1.3 ug/m3 of CO, NO2, and PM2.5, respectively) in Macao. For example, what was the domain of the CMAQ model used? Did it include a much wider region that encompassed other cities/provinces of China? What meteorological data and chemical schemes were used to run the model? What were the chemical and meteorological boundary conditions used to drive the CMAQ model? On Page 10, Lines 20-24, the authors mention turning off local stationary and mobile source emissions, but it is not clear what emissions inventory was used to drive the background concentrations of CO, NO2, and PM2.5 elsewhere. What about shipping emissions, which are sources of NOx? The emissions and meteorological data used to drive the 4 km x 4km CMAQ model need to be described in detail; the

CMAQ model is as critical as the AERMOD model in calculating the local vs. regional contribution. Since the prevailing wind direction is from the northeast (shown in Figure S5), it appears there would be a strong influence from emissions occurring in Hong Kong and other major cities in Southeast China.

(v) If the authors' used CMAQ to calculate background concentrations, why isn't CMAQ also used to quantify the local contribution of vehicle emissions to ambient concentrations of CO, NO2, and PM2.5, along with AERMOD? Some of these concerns I have raised with regards to chemistry could be mitigated with a chemical transport model like CMAQ. If there is similarity in the result between AERMOD and CMAQ in the local vs. regional contribution, then chemistry may not play such an important role and the modeling results presented may be valid.

Specific Comments

(2) Page 6, Lines 6-15: What were the criteria used that defined a "typical" road link? Especially, how were the 5 road links investigated for the entire day chosen? For example, in Figure S3, it appears that many of the observations were on arterial and residential roads, and relatively few observations on freeways. However, I would think it would be more important to characterize the freeways since they have much higher traffic volumes, and account for a significant fraction of vehicle traffic. Also, it would help to create a map similar to Figure S6, showing traffic volumes for each link simulated using the TransCAD model, and to also highlight which links were surveyed.

(3) Page 6, Lines 27-28: It would help to show a line with trucks in Figure S2 to illustrate this point.

(4) Page 7, Line 2: To extrapolate to other hours using Equation 3, how consistent is the temporal variability observed across road links? If they are consistent, then it is appropriate to spatially model traffic flows for the 6 PM hour only, and to extrapolate traffic patterns to other times of the day. However, if they are not, I would imagine it is better to run the traffic model for each hour of the day. To support the assumption

that temporal variability is consistent across space, Figures 2 and 3 would benefit from estimating uncertainty bands for each road class shown.

(5) Section 2.4: It is not clear from the description here of the EMBEV-Macao model whether gross-emitters are taken into account with the emission factor data collected between PEMS and remote sensing. Average emission factors could be significantly underestimated if gross-emitters are not included (Bishop et al., 2012). Also, how are cold start emissions taking into account? It is ok to reference prior papers, but I think it is important to address these issues explicitly here.

Bishop, G. A., et al. (2012), Multispecies remote sensing measurements of vehicle emissions on Sherman Way in Van Nuys, California, J. Air & Waste Management Association, 62.

(6) Page 9, Lines 17-18: It is important to describe here the advantages and disadvantages of Gaussian models in relation to the other types of models. As highlighted in Comment 1, I have concerns over whether Gaussian models can accurately model constituents that undergo complex chemistry, including NOx-VOCs-O3 and secondary aerosols, which are pertinent to this study.

(7) Page 14, Lines 20-24: The authors mention that traffic loop detector data is collected in many Chinese cities. Is traffic loop detector data not being collected in Macao? If so, it should be mentioned here.

(8) Section 3.3: The locations of the ambient monitoring locations should be shown on a map somewhere (e.g., Figure S1).

(9) Figure 8: Why are results not shown for CO? It seems relevant to the model evaluation described (Page 15, Lines 29-31).

(10) Page 16, Lines 16-20: Another source of uncertainty are effects due to chemistry (see Comment 1).

(11) Table 4. For the most part, the fleet-averaged emission factors seem reasonable,

except for MDPV-Gasoline and LDT-Gasoline. Why are emission factors for CO and THC nearly as large as motorcycles, presumably with two-stroke engines, which are expected to have the highest emission factors for these pollutants?

(12) Tables 5 and 6. Too many significant figures are shown, especially for CO2. Probably no more than 3 significant figures are justified given uncertainties in emission intensities.

(13) Figure S5. Where are weather stations located? Should be shown on Figure S1.

Minor Comments (14) Page 5, Line 1: I believe there is a mistake here, that "vehicle classification f" should read "vehicle classification v".

(15) Page 7, Line 23: Better to report amount of data in hours collected rather than seconds; as a reader it is hard to comprehend how much data was collected using the latter units.

(16) Page 15, Line 22: I believe there is a mistake here, "Table 6" should read "Table 7".

(17) City boundaries shown in Figure 7 and Figure S7, are hard to see. Suggest darkening the boundaries.

(18) Figure S2. The vertical axis labeling is confusing. Instead of ratios, I think fraction of total traffic counts better describes what is being shown.

---

## Author Comment (AC1) · 27 Jun 2016

Please find our detailed responses to the referees' comments in the uploaded package. In addition, we also attach a manuscript with changes highlighted in yellow and the revised supplementary information.

Please also note the supplement to this comment:
http://www.atmos-chem-phys-discuss.net/acp-2016-69/acp-2016-69-AC1-supplement.zip

---

## Author Response (AR1)

Reply to comments on "The Challenge to NO$_X$ Emission Control for Heavy-duty Diesel Vehicles in China" by S. Zhang et al

*"Black" means the comments from reviewer and "Blue" text are our responses.*

We are deeply grateful to the referees of this paper for the very helpful comments. These comments are fully understood by the authors and individually responded to. Our responses to the comments are listed below.

**Reply to comments from Anonymous Referee #1:**

**Overall Comments:** My overall comments are favorable. I thought the paper presented interesting results of the use of a vehicle emission model to infer the dominant sources of vehicle pollution in Macao, as well as the potential use of traffic information linked to an emission model and air dispersion model to inform environmental and transportation policy. My major comment on the content of the paper, is that the authors need to provide more information into how the emissions and air quality concentrations were estimated are estimated. For example, the paper is unclear on how the vehicle split by link is estimated, how the speeds are estimated and applied in the vehicle emission model, and how the 'fleet average' emission rates in Table 4 are estimated. Also the comment Printer-friendly version Discussion paper discussion on the air dispersion modeling is very limited.

We appreciate the referee's favorable comments. According to specific comment on the method and data details, we try out best to inform the readers of the estimation of fleet split and speed profiles (e.g., Equations 3, 5 and 6, see Page 7), the development of local emission factors (e.g., Page 9 to Page 10). In the revised manuscript, we use the gasoline LDPVs as example to illustrate how to develop the localized emission factors based on measurement data. In the Table 4, we also note the emission measurement data sources for each fleet.

Furthermore, according to the second referee's comment. We acknowledge the limitation of NO$_2$ concentration simulation by using the AERMOD model (e.g., lack of adequate high resolution O$_3$ concentration profiles, simplified chemical reaction mechanism) (see Page 12, Lines 1 to 11). Therefore, we add a discussion to note the research requirement for high-resolution air quality modeling with detailed model configurations of the CMAQ system (see the Supplementary Information, and Page 18 Line 29 to Page 19 Line 12).

I would also strongly recommend the authors change the wording on page 2 (line 25) and page 17 (line 6) from 'irreplaceable' tool, to 'can be to a valuable tool'. I do not think the paper showed that the high-resolution traffic tool is an irreplaceable assessment tool. The paper did show that the results from the tool, appear to compare reasonably well with at least one air quality monitor, and it discusses ways, in which it could be used to inform air quality and policy decisions in the future.

Page 4. Line 4, huge a large transportation demand.

We revise these wordings according to these two comments.

Page 6. Line 21. What is the MC fraction on the Macao Peninsula?

The average observed MC fraction is approximately 45% compared with the 35% for LDPVs, as we note in the revised manuscript (see Page 6, Lines 6 to7).

Page 7, Line 24. You should mention the variability in the speed trends across roadway links which could be due to the limited data based on chase-car study.

Figure 3 appears to have significant variability from hour to hour, that I would think would be smoother if it had a larger sample size across more sample days, more links, and more vehicles. This should at least be discussed, especially if confidence intervals of the mean speed are not presented (which I think would show that many of the hourly mean speeds are not significantly different than one another).

We thank the referee for this kind suggestion. In the revised manuscript, we first in detail illustrate the equation to map the speeds (see Equation 3 in Page 7, and Equation 6 in Page 8), and then note the uncertainty from the area-aggregated data at the hourly and link level. For example, we now report the coefficients of variation for hourly speeds of arterial roads in the MP during three different hours (e.g., approximately 40% to 50%) (see Page 8, Lines 23 to 29), and estimated the effect on $CO_2$ emission factors for gasoline LDPVs as a case (see Page 17, Lines 13 to 18). We also recommend the useful application of ITS approach to capture the real-world variability of traffic dynamics.

Page 11. How do you obtain estimates of vehicle classifications by link? This is not clear to me from reading the first paragraph on page 11.

In the method section, we add the Equation 5 in Page 7 to clarify the issue.

Page 14. Line 8-9. I think you mean higher emission rates, for lower level of service?

Page 14. Line 17. 'broad' instead of 'board' Page 14.

Page 14. Line 25-26. Rephrase this sentence. 'poor representativeness. . ..'

Revisions are done according to the referee's comment.

Page 15. Line 1-3. How does the daily variations in speeds, results in the variation in $CO_2$ emission factors? Is that from analysis done from the Beijing study? Or is that variation in link speeds applied to your emission model for Macao? Please be clear.

First, we clarify that the variation in $CO_2$ emissions is estimated for Beijing in the revised manuscript.

Second, for the referee's information, we applied PEMS testing profiles of 41 gasoline cars (16 in Macao, 11 in Beijing and 14 in Guangzhou) to establish the "real-world" speed correction function for $CO_2$ emission factors (similar works also done for other vehicle groups and pollutant species) (Zhang et al., 2014). It is noted that we improve the extraction of speed effects by constructing the baseline emission factor using the operating mode method (i.e., speed and VSP binning). Thus, we gain the speed correction function with regression correlation coefficient ($R^2$) higher than 0.9, and the uncertainty range of -20%/+13 at a 95 confidence level (average speed lower than 60 km $h^{-1}$). In addition, we have not observed significant difference between the speed effects among various cities. The speed correction can be well applied from link level to trip/road network level with relative bias of -13%/+11%.

Zhang, S., Wu, Y., Liu, H., Huang, R., Un, P., Zhou, Y., Fu, L., Hao, J.: Real-world fuel consumption and $CO_2$ (carbon dioxide) emissions by driving conditions for light-duty passenger vehicles in China. Energy, 69, 247-257, 2014.

Page 16. Line 5-9. Rewrite sentence, and improve grammar

Page 16. Line 29. Start new paragraph.

Page 17. Line 6. Suggest 'can be a valuable assessment tool' (not ' irreplaceable')

Page 17. Line 17. Replace 'significantly less traffic' with 'smaller' Page 17. Line 30.

Revisions are done for the comments above.

The Taxis are diesel powered? This should be clarified in the main text, as well as in Table 4. RE: Table 4. I am surprised that the diesel Taxis' have lower NOx g/km, than the MDPV gasoline vehicles? Are these emission rates based on PEMS data or emission standards? If some of these emission rates are based on certification vehicle standards, than the paper should mention the uncertainty of using vehicle emission standards (particularly EURO diesel standards) to represent real-world emission rates. Also, similarly, why is the MDPV diesel in the same range as the MDPV gasoline vehicles? In general, more information is needed on the derivation of the fleet-average emission factors in Table 4.

First, all the taxis in Macao are powered by diesel. The gaseous emission rates for diesel taxis are developed largely based on local PEMS data (Hu et al., 2012).

Second, we don't have dynamometer or PEMS testing data for MDPV-gasoline vehicles. High MDPV-gasoline emission factors are because we applied the emission parameters in our previous EMBEV model (Beijing). Therefore, in the revised manuscript, we revised the emission factors for MDPV-gasoline as well as LDT-gasoline based on the remote sensing based results (i.e., fuel based emission ratios between MDPV-gasoline to LDPV-gasoline). The revised emission factors are presented in the Table 4. These modifications would lead to lower total vehicular emissions of THC by 1%, CO by 4% and $NO_X$ by 1%, which would play a minor role in the overall temporal and spatial emission patterns. We have revised the data and figure throughout the manuscript according to the new emission factors for MDPV-gasoline and LDT-gasoline (see Table 4).

In addition, because no emission standards have been adopted in Macao until 2012, therefore, the emission standard category defined by the emission model actually represents the aggregated model year group for emission estimation. The uncertainty in fleet-average emission factors is related to the data availability (sample size) and fleet configuration. For example, for diesel taxis that were dominant by one vehicle model (e.g., Toyota Corolla diesel), the relative uncertainty ranges (95% CI) of average emissions are 37% for THC, 22% for CO and 48% for $NO_X$. The uncertainty for MDPV-gasoline and LDT-gasoline would be higher because of less data samples. However, the traffic fractions for MDPV-gasoline and LDT-gasoline are less than 2.5%, so their impacts on total vehicle emissions are less significant.

**Reply to comments from Anonymous Referee #1:**

This study develops a high-resolution motor vehicle emissions inventory for a city in China, models the inventory, and evaluates the modeling results against ambient monitoring data. Main findings of this paper are that it is important to capture the spatial heterogeneity of the vehicle fleet mix across the urban domain, and to use local information on emission factors. Overall, the authors present a novel approach to mapping vehicle emissions, especially in cities where traffic activity and emission factor data are not as readily available. This is a major accomplishment and worth replicating in other cities. In regards, to the second major aspect of this study, the air quality modeling, I have some concerns that I believe need to be addressed more fully in revision. My comments mostly refer to the treatment of atmospheric chemistry in the dispersion model. With major revision, I do believe it is possible for this manuscript to be considered for publication in Atmospheric Chemistry & Physics.

We appreciate the referee's positive comments on traffic and emission aspects. In the initial manuscript, we attempted to use the $NO_2$ simulation results as a validation of emission inventory results. After carefully considered the comments from this referee, we acknowledge the model limitation of the application in the city scale. In addition, the uncertainty from air quality modeling may also undermine the efforts of system validation.

Therefore, in the revised manuscript, we additionally use the statistical fuel consumption data to validate emission inventory, which is feasible because Macao is a specially closed island city (i.e., one Special Administration Region of China that requires special certificates for cross-border vehicle use) (see Page 14 Line 28 to Page 15 Line 15). The results indicate that a nice agreement between gasoline consumption record and gasoline fuel $CO_2$ emissions. Although this might be still not sufficient to address emissions more related to diesel fleets (e.g., $NO_X$, $PM_{2.5}$), we could see it as a robust evaluation of overall traffic patterns by avoiding the uncertainty of air quality modeling.

Second, we note the major limitations of the $NO_2$ concentration simulation with the AERMOD in the methodology section, which include the absence of other $NO_X$ related reactions, the lack of adequate ozone concentration profiles, and the simplified time framework of $NO/NO_2$ conversion (see Page 12 Lines 1 to 18). The $NO_2$ concentration simulation with the AERMOD is not included in the revised manuscript. However, given the fact of $NO_2$ pollution in Macao, we add a discussion on the future research requirement of high-resolution air quality modeling, because the CMAQ model might underestimate the $NO_2$ concentration for traffic populated areas and shrink the useful features of link-level emission inventory (see Page 18 Line 28 to Page 19 Line 11, and the Supplementary Information). The following comments regarding the $NO_2$ concentration simulation are also responded individually.

We appreciate the papers this referee suggested, which are helpful to understand the complex chemical transport and vehicle emissions. We also adequately add these publications as references for readers' information.

General Comments (1) My concerns with respect to the air quality modeling are with the treatment of chemistry, and how background levels are estimated. Unless the following concerns can be addressed, I believe that statements that quantify the fractional contribution of motor vehicle emissions to ambient concentrations observed should be removed (bottom of page 15 and top of page 16), and commentary restricted to qualitative statements.

We have removed this paragraph from the manuscript. In addition, we add the detailed setups of the CMAQ modeling in the supplementary information, which supports the discussion on the research requirements for high resolution air quality modeling.

(i) More detail is needed on how a dispersion model like AERMOD accounts for chemistry, especially for NO2. Given that authors present high-resolution air quality maps, it seems important to capture spatial gradients that may arise due to interactions between ozone and fresh NO emissions; i.e., ozone tends to be suppressed near highways, and NO2 elevated (Murphy et al., 2007). On page 10, Lines 15-17, the authors mention using ozone data to account for the oxidation of NO to NO2, but do not describe how. How many monitoring sites are used in this calculation? Where are they located, and what is their proximity to roadways? What is the timescale of the NO -> NO2 conversion employed in AERMOD, and how was this estimated from observations?

Murphy, J. G., Day, D. A., Cleary, P. A., Wooldridge, P. J., Millet, D. B., Goldstein, A. H., and Cohen, R. C.: The weekend effect within and downwind of Sacramento – Part 1: Observations of ozone, nitrogen oxides, and VOC reactivity, Atmos. Chem. Phys., 7, 5327-5339, doi:10.5194/acp-7-5327-2007, 2007.

We do not have spatially resolved ozone profiles in Macao to support a dedicated simulation of NO$_2$ concentration. Ozone concentrations at three air quality monitoring sites (i.e., one in the MP, one in Taipa, and one in Coloane) were used as input for each region. The AERMOD model simply assumes that the oxidation of NO is instantaneous and irreversible on hourly basis. In the revised manuscript, all these limitations have been noted in the methodology section.

(ii) It is also not clear how AERMOD treats the loss of NO2 to PAN and HNO3. In an urban mass, these products of NO2 can comprise up to half of daytime NOy (= NOx +PAN + HNO3, see Pollack et al., 2012). If the authors' only account for the production of NO2 from fresh NO emissions, without accounting for the loss of NO2 from daytime chemistry, then the NO2 concentrations simulated from local vehicle emissions shown in Figure 7 could be overestimated. Consequently, the authors may overestimate the motor vehicle contribution to ambient NO2. The importance of the loss term will depend on the photochemical age of the air mass that reaches the monitoring site, which could vary by time of day, wind speed/direction, and synoptic events. A robust estimation of the local vehicle contribution to ambient NO2 (as shown in Figure 8) should take into account both the production and loss of NO2.
Pollack, I. B., et al. (2012), Airborne and ground-based observations of a weekend effect in ozone, precursors, and oxidation products in the California South Coast Air Basin, J. Geophys. Res., 117, D00V05, doi:10.1029/2011JD016772.
Two EPA Tier-3 methods (OLM and PVMRM) incorporated to the AERMOD both only take account of the oxidation of fresh NO to NO2. The other oxidation processes to NO$_Y$ products (e.g., HNO$_3$, PAN, NO$_3$) are ignored by the AERMOD. This issue has been noted in the manuscript (see Page 12 Lines 1 to 19).

(iii) As I understand, the NO$_2$ observations shown in Figure 8, are daily concentrations from a single monitoring site (Page 16, Lines 10-13). Is this the average of 24-hours of data? The use of daily averages could be influenced by nighttime chemistry, which presumably would not be taken into account with AERMOD. To avoid these complications, it is better to restrict the model comparison to daytime values only.

The original results are average of 24-hours data. This section is dropped off due to the model limitations.

(iv) I found the description of the CMAQ model (on Page 10, Lines 20-26) used to estimate regional background and cross-boundary transport lacking. This is important since background levels (Page 10: 304 ug/m3, 27 ug/m3, and 23 ug/m3 of CO, NO2, and PM2.5, respectively) are as big or much larger than the motor vehicle contribution to these pollutants (Page 15: 88 ug/m3, 22 ug/m3, and 1.3 ug/m3 of CO, NO2, and PM2.5, respectively) in Macao. For example, what was the domain of the CMAQ model used? Did it include a much wider region that encompassed other cities/provinces of China? What meteorological data and chemical schemes were used to run the model? What were the chemical and meteorological boundary conditions used to drive the CMAQ model? On Page 10, Lines 20-24, the authors mention turning off local stationary and mobile source emissions, but it is not clear what emissions inventory was used to drive the background concentrations of CO, NO2, and PM2.5 elsewhere. What about shipping emissions, which are sources of NOx? The emissions and meteorological data used to drive the 4 km x 4km CMAQ model need to be described in detail; the CMAQ model is as critical as the AERMOD model in calculating the local vs. regional contribution. Since the prevailing wind direction is from the northeast (shown in Figure S5), it appears there would be a strong influence from emissions occurring in Hong Kong and other major cities in Southeast China.

(v) If the authors' used CMAQ to calculate background concentrations, why isn't CMAQ also used to quantify the local contribution of vehicle emissions to ambient concentrations of CO, NO2, and PM2.5, along with AERMOD? Some of these concerns I have raised with regards to chemistry could be mitigated with a chemical transport model like CMAQ. If there is similarity in the result between AERMOD and CMAQ in the local vs. regional contribution, then chemistry may not play such an important role and the modeling results presented may be valid.

We add a section in the Supplementary Information to illustrate the regional air quality modeling with the CMAQ model. First, we did include a wider region in the CMAQ modeling framework, as a triple-nested simulation domain was applied. Domain 1 covers most of China of 36 km × 36 km horizontal resolution. Domain 2 covers East of China with 12 km × 12 km horizontal resolution. Domain 3 covers Perl River Delta (PRD) with 4 km × 4 km horizontal resolution. Second, in terms of emission input data, we referred to Zhao et al. 2013a and 2013b for the emissions in other provinces of China. The local emissions for other sectors (e.g., residential, power, and industrial sectors) in Macao were provided by the Macao Environmental Protection Bureau, together with the vehicle emissions estimated by this study. It is noted that the shipping emissions were not estimated by the local stakeholders due to the lack of ship position information (i.e., not in the local waters).

Although the CMAQ model is more sophisticated in chemical transport mechanisms than the AERMOD model, however, there are still significant limitations. First, the number of 4 km x 4 km cells (note: 2 cells only occupied by Macao, and 4 cells occupied by Macao and Zhuhai together, which is the city adjacent to Macao) are quite rare to cover the entire Macao, which indicates less spatial resolution. Second, the simulated results using the CMAQ is much lower than observed levels. Although the AERMOD model may yield higher NO2 concentrations in traffic populated areas, however, the model limitations would bring in considerable uncertainty (e.g., diurnal fluctuations). Thus, we suggest that future efforts are required to develop more advanced air quality model to enhance spatial heterogeneity and chemical transport at the same time. We add a discussion paragraph in the manuscript to highlight the research gap.

We clarify that CO was not included in the CMAQ modeling study (we checked this issue with the researcher who operated the regional air quality modeling). This was because the regional emission inventory did not report results for CO. The regional background of ambient CO concentration was approximated by the CO concentration of an air quality monitoring station in the remote rural area of Hong Kong (air quality data in the Mainland China were not publicly available then).

Zhao, B.; Wang, S. X.; Dong, X. Y.; Wang, J. D.; Duan, L.; Fu, X.; Hao, J. M.; Fu, J., Environmental effects of the recent emission changes in China: implications for particulate matter pollution and soil acidification. Environmental Research Letters 2013a, 8, (2).

Zhao, B.; Wang, S.; Wang, J.; Fu, J. S.; Liu, T.; Xu, J.; Fu, X.; Hao, J., Impact of national NOx and SO$_2$ control policies on particulate matter pollution in China. Atmospheric Environment 2013b, 77, (0), 453-463.

(2) Page 6, Lines 6-15: What were the criteria used that defined a "typical" road link? Especially, how were the 5 road links investigated for the entire day chosen? For example, in Figure S3, it appears that many of the observations were on arterial and residential roads, and relatively few observations on freeways. However, I would think it would be more important to characterize the freeways since they have much higher traffic volumes, and account for a significant fraction of vehicle traffic. Also, it would help to create a map similar to Figure S6, showing traffic volumes for each link simulated using the TransCAD model, and to also highlight which links were surveyed.

First, the tropology of road network in Macao is significantly different with that in other large cities in Mainland China or the US. Because of very densely populated city landscape, the total number and length share (e.g., 12%) of urban freeways in Macao are less than other larger cities. These urban freeways in Macao are all three cross-sea bridges or the main traffic corridors connected to these bridges. We agree with the reviewer's comment that urban freeways should be paid more attention to. In this study, we investigated the traffic volume data for six urban freeway links, which accounted for 17% of the total length for urban freeways in Macao. This investigation proportion is higher than that for arterial roads and residential roads (both less than 10%). The 5 typical roads were selected according to their road class and region (now noted in the revised manuscript), including one urban freeway, two arterial roads, two residential roads (see Page 6 Lines 9 to 13). The road links used in the GIS map are highly fragmented (e.g., average link length below 200 m for arterial and residential roads) because of the densely distributed and intersected roads, which also lead to lower proportions of the coverage. This issue is also the main reason to apply the TransCAD for volume mapping. We add a new figure (Fig. S2) in the Supplementary Information to highlight the links with observation data.

(3) Page 6, Lines 27-28: It would help to show a line with trucks in Figure S2 to illustrate this point.

We have revised the current Fig. S3 by adding the hourly volume fractions for trucks (LDTs and HDTs combined). In two regions (e.g., MP and TCC), the hourly volume fractions for trucks in the total fleet were both higher during daytime, and a major part was contributed by LDTs (~70% of total trucks in daytime).

(4) Page 7, Line 2: To extrapolate to other hours using Equation 3, how consistent is the temporal variability observed across road links? If they are consistent, then it is appropriate to spatially model traffic flows for the 6 PM hour only, and to extrapolate traffic patterns to other times of the day. However, if they are not, I would imagine its better to run the traffic model for each hour of the day. To support the assumption that temporal variability is consistent across space, Figures 2 and 3 would benefit from estimating uncertainty bands for each road class shown.

We add the standard deviations for hours from 6 a.m. to 11 p.m. in the Fig. 2 for each road category to indicate the traffic volume bias among individual roads. To improve the presentation quality, we split the figure into three sub-figures. Nevertheless, the wide bias of hourly traffic volume data are largely attributed to the variations in designed capacity (e.g., number of lanes), location and other issues. As for Fig. 3, adding the uncertainty ranges would be very occupied by too many makers in one figure. Thus, we state the spatial bias of hourly speeds during various hours and estimate the impact on the variability of emission factors (see Page 8 Lines 24 to 27).

Therefore, to better watch the consistency of the temporal variability between individual roads, we estimate the average hourly allocation of traffic volumes for the period from 6 a.m. to 11 p.m (see the figure below, added as Fig. S5). The results indicate that the average correlation variations (i.e., the ratio of standard deviation to mean value) are 13% for freeways, 14% for arterial roads, and 16% for residential roads (note: observed data only, not including links without field investigation). Therefore, given the narrow relative bias regarding the temporal variability of traffic volume data, we use the traffic volume for 6 p.m. to estimate other times of the day as an efficient way, because running traffic demand models would also bring in uncertainty. We add this required information in the revised manuscript.

[Figure]

[Figure]

[Figure]

Figure S5. Average allocations of hourly traffic volume in the total traffic volume from 6 a.m. to 11 p.m. Only roads with observed traffic volume data included in this figure.

(5) Section 2.4: It is not clear from the description here of the EMBEV-Macao model whether gross-emitters are taken into account with the emission factor data collected between PEMS and remote sensing. Average emission factors could be significantly underestimated if gross-emitters are not included (Bishop et al., 2012). Also, how are cold start emissions taking into account? It is ok to reference prior papers, but I think it is important to address these issues explicitly here.

Bishop, G. A., et al. (2012), Multispecies remote sensing measurements of vehicle emissions on Sherman Way in Van Nuys, California, J. Air & Waste Management Association, 62.

According to the reviewer's comment, we revised this section with more clarifications about the emission factor development. We use the gasoline LDPV as an example to illustrate key processes. First, we used the remote sensing data to observe the long-term emission trends by model year and to develop several model year groups.

Second, for each model year group, we developed the emission parameters according to the local PEMS results with adequate modifications. It is noted that the original EMBEV model has developed distribution functions of individual emission factors based on large-sized vehicle samples (e.g., dynamometer tests, PEMS tests). We applied the function curve (long-tail distribution with the presence of high-emitters) to estimate the effect of high-emitters. Third, according to the original EMBEV framework, we modified the parameters regarding speed correction and start emissions (both using the PEMS data) and corrected other local features (e.g., fuel quality, environmental conditions). For some fleets that have few tests data involved in the EMBEV model, we developed the emission factors based on the remote sensing results (e.g., motorcycles) (see Page 9 Line 28 to Page 10 Line 21).

In the Table 4, we now have added the data sources for readers' better information.

(6) Page 9, Lines 17-18: It is important to describe here the advantages and disadvantages of Gaussian models in relation to the other types of models. As highlighted in Comment 1, I have concerns over whether Gaussian models can accurately model constituents that undergo complex chemistry, including NOx-VOCs-O3 and secondary aerosols, which are pertinent to this study.

Please see our associated comments above.

(7) Page 14, Lines 20-24: The authors mention that traffic loop detector data is collected in many Chinese cities. Is traffic loop detector data not being collected in Macao? If so, it should be mentioned here.

In Macao, the intelligent transportation system is not well developed. The traffic loop, floating car system (based on taxis), and radio frequency identification detectors are not present in Macao. We note this in the revised manuscript (see Page 8 Lines 27 to 29; Page 17 Line 28).

(9) Figure 8: Why are results not shown for CO? It seems relevant to the model evaluation described (Page 15, Lines 29-31).

As we have clarified in a previous response, regional CO concentration was not simulated by the CMAQ model. Therefore, we are not able to adequately use CO concentration to evaluate the model.

(10) Page 16, Lines 16-20: Another source of uncertainty are effects due to chemistry (see Comment 1).

Please see our response to the Comment 1 above.

(11) Table 4. For the most part, the fleet-averaged emission factors seem reasonable, except for MDPV-Gasoline and LDT-Gasoline. Why are emission factors for CO and THC nearly as large as motorcycles, presumably with two-stroke engines, which are expected to have the highest emission factors for these pollutants?

As our response to the first referee, we don't have dynamometer or PEMS testing data for these two vehicle categories (e.g., LDT-Gasoline, MDPV-Gasoline) in Macao. We understand the concerns from the referee and revise the emissions by refereeing to the local remote sensing results. To be specifically, we now estimate their fleet-average emissions based on the ratios of their average fuel-based emissions to those of LDGVs. For example, the remote sensing results indicate that fuel-based emissions of THC, CO, and $NO_X$ are higher than LDPV-Gasoline by 285%, 172%, 132%, respectively, although the average engine size of the LDT-gasoline is smaller than that of LDPV-gasoline. So, fleet-average emission factors of LDT-gasoline for CO, THC and $NO_X$ are revised as 6.36 g km$^{-1}$, 1.75 g km$^{-1}$ and 0.61 g km$^{-1}$ (See Table 4). High emission factors may attributed to relatively poorer usage and maintenance conditions of the LDT-gasoline for freight purpose than those of LDPVs mainly for passenger transportation. Changes for MDPV-gasoline are made in a similar way. It is noted that compared with LDPV-gasoline vehicles, the numbers of valid remote sensing samples for LDT-gasoline and MDPV-gasoline are both significantly less, indicating potentially higher uncertainty in emission factor results. However, on the other hand, the total traffic volume fractions for LDT-Gasoline and MDPV-gasoline are both less than 2.5%, so the variations in emission factors would only lead to minor variations in the total emissions (1% for CO and 4% for $NO_X$).

In terms of the estimated emission factors for MC, the reviewer understands correctly. The significantly higher THC emission factors for MC-light are because of two-stroke engines, and the deterioration is very significant according to the remote sensing data by model year (Zhou et al., 2014). Based on the remote sensing results, both MC-light (two-stroke) and MC-heavy (four-stroke) have higher CO emissions than gasoline passenger cars.

Zhou, Y., Wu, Y., Zhang, S., Fu, L., Hao, J.: Evaluating the emission status of light-duty gasoline vehicles and motorcycles in Macao with real-world remote sensing measurement. J. Environ. Sci., 26(11): 2240-2248, 2014.

(12) Tables 5 and 6. Too many significant figures are shown, especially for CO2. Probably no more than 3 significant figures are justified given uncertainties in emission intensities.

We revise the data presentation for emissions by limiting significant figures less than three.

(13) Figure S5. Where are weather stations located? Should be shown on Figure S1.

We now have marked the location of weather stations.

Minor Comments (14) Page 5, Line 1: I believe there is a mistake here, that "vehicle classification f" should read "vehicle classification v".

Revision is done.

(15) Page 7, Line 23: Better to report amount of data in hours collected rather than seconds; as a reader it is hard to comprehend how much data was collected using the latter units.

Revision is done by using hour as the unit.

(16) Page 15, Line 22: I believe there is a mistake here, "Table 6" should read "Table 7".

Revision is done.

(17) City boundaries shown in Figure 7 and Figure S7, are hard to see. Suggest darkening the boundaries.

We improve these figures according to the reviewer's suggestion.

(18) Figure S2. The vertical axis labeling is confusing. Instead of ratios, I think fraction of total traffic counts better describes what is being shown.

Revision is done.

[revised manuscript text omitted]